# Artificial Combined Model Based on Hybrid Nonlinear Neural Network Models and Statistics Linear Models—Research and Application for Wind Speed Forecasting

**Yuewei Liu [1], Shenghui Zhang [1,\*], Xuejun Chen [2] and Jianzhou Wang [3]**

[1]  School of Mathematics and Statistics, Lanzhou University, Lanzhou 730000, China; lyw@lzu.edu.cn
[2]  Gansu Meteorogical Service Centre, Lanzhou 730020, China; xuejunchen1971@163.com
[3]  School of Statistics, Dongbei University of Finance and Economics, Dalian 116025, China; wjz@lzu.edu.cn
\*  Correspondence: zhangshh17@lzu.edu.cn; Tel.: +86-186-9872-7162

**Abstract:** The use of wind power is rapidly increasing as an important part of power systems, but because of the intermittent and random nature of wind speed, system operators and researchers urgently need to find more reliable methods to forecast wind speed. Through research, it is found that the time series of wind speed demonstrate not only linear features but also nonlinear features. Hence, a combined forecasting model based on an improved cuckoo search algorithm optimizes weight, and several single models—linear model, hybrid nonlinear neural network, and fuzzy forecasting model—are developed in this paper to provide more trend change for time series of wind speed forecasting besides improving the forecasting accuracy. Furthermore, the effectiveness of the proposed model is proved by wind speed data from four wind farm sites and the results are more reliable and accurate than comparison models.

**Keywords:** modified cuckoo search algorithm; combined model; hybrid nonlinear models; wind speed forecasting

## 1. Introduction

In the past few decades, due to policy-driven environmental and energy security issues, the development of renewable energy sources (RESs), which play an indispensable role in the global power sector, has received much attention, and these energy sources play an indispensable role in the global power sector [1].

Wind energy is a very rich resource on the earth, according to a report of the World Meteorological Organization (WMO). Global wind energy reserves total about $2.74 \times 10^9$ MW, which amounts to wind energy development and utilization of approximately $2 \times 10^7$ MW, greater than hydrogen energy, and can be developed and utilized all around the world. It is estimated that the amount of wind energy is 10 times larger than hydrogen energy, and Earth's daily wind power is equivalent to the current world energy consumption. Almost all of the world's energy is produced by coal combustion, and only one-third is provided by wind energy [2].

Therefore, accurate prediction of wind speeds over several periods, from a few minutes (short-term) to a few hours (medium-term) or days (long-term), is a strategic issue that needs to be addressed. Forecasting short-term wind speed is key to improving the reliability of wind power generation systems and integrating wind energy into the power grid [3–7]. In order to reduce the operational cost of a wind farm, it is very important to improve short-term wind speed forecasting accuracy.



However, wind power generation has some drawbacks. One of the main problems is that wind is an intermittent energy source, which means that there are great differences in the production of energy due to various factors, such as wind speed, air density, and wind turbine characteristics. Another problem is that wind is often used as a nondispatchable source of energy, so the management of wind energy production according to demand is difficult. Usually, intermittency can be considered as a problem related to dispatchability [8]. According to the theory of wind energy, wind energy is proportional to wind speed, so seemingly trivial forecasting changes in wind speed might lead to significant changes in total wind power.

In this condition, wind speed forecasting is the significant foundation and premise. More accurate forecasting of wind speed can: (1) reduce the rotation of wind farm equipment and operating costs, (2) improve the wind power penetration limit, (3) help scheduling in a timely manner to adjust plans, (4) reduce the impact of wind on the grid, and (5) effectively reduce or avoid the negative impact of wind farms in the power system, improving the ability of wind power in the competitive electricity market.

It is a difficult task to develop applications and methods of wind speed forecasting. In order to solve wind speed forecasting problems, most researches are based on terrain features, atmospheric pressure, ambient temperature, and other meteorological information to obtain a satisfy results. However, wind speed, which is generally considered to be one of the most difficult weather parameters, can be forecasted based on its chaotic and random fluctuations [9–11]. In order to achieve accurate wind speed forecasting, many methods and models have been proposed, which can be grouped into four categories [11]: (a) statistical models, (b) physical models, (c) artificial intelligence models, and (d) spatial correlation models.

Statistical models used to forecast wind speed usually are fuzzy methods [12], autoregressive (AR), moving average (MA), autoregressive moving average (ARMA) [13], and autoregressive integrated moving average (ARIMA) [14] models. These required that the time series data be stable, or stable after differentiation, and they can only capture the linear relationship, but not the non-linear relationship. So, statistical models are not suit for nonlinear data.

Physical methods can be used to forecast simulated wind speed, but have a great influence on the results of numerical simulation for prediction accuracy. Physical methods based on large samples, the forecast object area of pressure, temperature, terrain, obstacles, and other objective conditions have higher requirements than numerical prediction methods in the prediction of wind speed, and systematic errors often arise due to the physical parameters and the processing of grids. One way to solve this problem is to improve the grid precision of the numerical prediction method where calculation accuracy is limited by grid resolution. Higher grid resolution not only takes a lot of computing time, but also is beyond the assumptions of a mesoscale model, and an improper boundary layer parameterization scheme may lead to larger calculation errors. The model itself also has some inherent defects, such as mode, near-ground wind speed with too-rapid height increase, and the wind speed forecast is not accurate [15–17].

Artificial neural network [18–28] has strong learning and mapping ability and can easily fit the arbitrary complex nonlinear relationship, which is very suitable for short-term wind speed forecasting, and now research with neural networks is quite active in the world. Commonly, researchers forecast wind speed using back-propagation neural networks (BPNNs). Usually wind direction, temperature, pressure, and other meteorological factors related to wind speed are applied as the input of the neural network and the wind speed value is the output of the sample. Training the network can establish a nonlinear simulation of wind speed and forecast future wind speed change. However, there is usually no fixed theory and it needs to rely on experience to guide the design of the network structure. In addition, selecting the appropriate training sample is also a difficult problem. When the model is training the samples, it is easy for the neural network to fall into a local minimum, or the fitting for the training sample reduces the network generalization ability, influencing the forecasting effect. Artificial

neural networks have powerful nonlinear mapping capabilities, and the modeling process is more concise, straightforward, and direct than other models.

Unlike other models, the spatial correlation model considers not only the wind speed of the given wind farm, but also the wind speed at several adjacent locations. Spatial correlation and the spatial correlation method need to consider the wind and wind speed data of several adjacent sites by using the spatial correlation between these sites to forecast wind speed. Due to wind speed data, the spatial correlation method requires multiple sites and the amount of data is very large, but also the requirement of real-time data acquisition and transmission is higher, and the current study is not very mature.

In practice, a variety of forecasting models can be chosen to forecast confirmed variables. Different forecasting models provide different information and the forecasting accuracy is often different. If some forecasting models with larger errors are simply discarded, some useful forecasting information would be lost, which is a waste of forecasting information and should be avoided. While existing time series forecasting models for researching the description and prediction of accuracy have reached a higher level, any kind of model is a simplified abstraction of the actual object, incomplete and with unavoidable limitations. A more scientific way is to combine different forecasting models into new combined models, to find a model based on the single model and optimization algorithm, or to combine time series forecasting models, as combined predictions can more fully reflect the dynamic phenomenon of internal regularity and future trends, thus constructing a combined forecasting model on the basis of single models.

Combining forecasting models is a comprehensive utilization of all kinds of forecasting models, built in the form of an appropriate combined model to forecast the variable. Combined forecasting models can avoid the loss of information in fitting a single model, reduce the randomness, and improve forecasting precision [29]. The main purpose of the method is comprehensive utilization of various models of the information provided, to improve forecasting accuracy as much as possible, and the greater comprehensiveness can be reflected by the variety of regulation of the system.

A new combined model that integrates nonpositive constraint theory [30], two hybrid neural networks, a single nonlinear model, two linear models, and a modified cuckoo search algorithm with steepest descent (we call it SDCS) is proposed in this paper, and 10-min time series wind speed data from four wind farm sites are applied to examine our proposed model. As a result of our experiments and analyses, the combined model has better performance than the other two hybrid models and three single models. Hence, the proposed wind speed forecasting combined model is helpful for wind energy utilization rates, such as avoiding wasting wind energy resources, saving on economic dispatching, reducing production costs, and improving wind turbine safety operation. This model also has certain reference value for decision-making of wind farms in practice. The combined forecasting model with accuracy is a promising model for future applications and can be utilized in other forecasting fields.

**The major contributions and innovations of this paper are as follows:**

(1) Based on decomposing the time series into the sum of interpretable components such as trend, periodic components, and noise with no a priori assumptions about the parametric form of these components, the preprocessing technique reduces the uncertainty and irregularity of wind speed data, and effectively improves the performance of wind speed forecasting.

(2) A novel deciding weights method based on a modified cuckoo search algorithm with steepest descent (SDCS) is proposed to improve the weights and optimize the performance. Due to the excellent performance of SDCS, the combined model effectively utilizes the advantages of its component models and overcomes the disadvantages of low precision and poor stability of traditional models.

(3) Linear functions and nonlinear functions are applied to test the wind speed series. Based on the results of our tests, the wind speed data cannot be considered as linear data or nonlinear

data completely, so the linear models and nonlinear models are applied correctly in our proposed model.

(4) The effectiveness of the proposed model is proved by wind speed data of four wind farm sites and the results are more reliable and accurate than comparison models. Our proposed combined model includes a hybrid BPNN, which is optimized by a modified ant colony optimization algorithm; a hybrid extreme learning machine neural network (ELMNN), which is optimized by differential evolution (DE); two traditional linear statistics models (ARIMA and Holt–Winters); and a single fuzzy neural network model (adaptive network-based fuzzy inference system), which can obtain high accuracy and strong stability.

(5) This novel combined model provides powerful technical support for smart grid scheduling and management. From the results of our experiments based on wind speed data of four sites, accurate forecasting can be achieved to enhance the security and controllability of the power grid and to realize reasonable dispatch of the power grid.

The remainder of the paper is arranged as follows. Singular spectrum analysis, nonlinear back propagation, extreme learning machine neural network, adaptive neuro-fuzzy inference system, two linear models, autoregressive integrated moving average, and Holt–Winters, heuristic algorithms and the optimization procedure is introduced in Section 2. Section 3 shows the proposed integrated framework. In Section 4, some forecasting performance metrics and the forecasting results of individual models and of the proposed combined model and comparisons are discussed and the views and results of the full paper are summarized. Finally, Section 5 concludes the study.

## 2. Methods

In this section, the data denoising method, statistic models, artificial neural networks and optimization algorithms are presented in this part.

### 2.1. Singular Spectrum Analysis

Singular spectrum analysis (SSA) can be used as a denoising technique so that it can be applied to arbitrary time series, especially nonstationary time series. The basic aim of SSA is to decompose the time series into the sum of interpretable components such as trends, periodic components, and noise with no a priori assumptions about the parametric form of these components [31].

Consider a real-valued time series $X = (x_1, \ldots x_N)$ of length $N$. Let $L(1 < L < N)$ be some integer called the window length and $K = N - L + 1$.

1st step: Embedding.

Form the trajectory matrix of the series $X$, which is the $L \times K$ matrix

$$X = [x_1, \ldots x_N] = \left(x_{ij}\right)_{i,j=1}^{L,K} = \begin{bmatrix} x_1 & x_2 & x_3 & \cdots & x_K \\ x_2 & x_3 & x_4 & \cdots & x_{K+1} \\ x_3 & x_4 & x_5 & \cdots & x_{K+2} \\ \vdots & \vdots & \vdots & \ddots & \vdots \\ x_L & x_{L+1} & x_{L+2} & \cdots & x_N \end{bmatrix} \tag{1}$$

where $X = (x_1, \ldots x_{i+L-1})^T, (1 < i < K)$ are lagged vectors of size $L$. The matrix $X$ is a Hankel matrix which means that $X$ has equal elements $x_{ij}$ on the anti-diagonals $i + j = const$.

2nd step: Singular Value Decomposition.

Perform the singular value decomposition ($SVD$) of the trajectory matrix $X$. Set $S = XX^T$ and denote by $\lambda_1, \ldots, \lambda_L$ the eigenvalues of $S$ taken in the decreasing order of magnitude $\lambda_1 \geq \ldots \geq \lambda_L \geq 0$ and by $U_1, \ldots, U_L$ the orthonormal system of the eigenvectors of the matrix $S$ corresponding to these eigenvalues.

Set $d = rankX = max\{i, such\ that\ \lambda_i > 0\}$ (note that $d = L$ for a typical real-life series) and $V_i = X^T U / \sqrt{\lambda_i}, (i = 1, \ldots, d)$. In this notation, the SVD of the trajectory matrix $X$ can be written as $X = X_1 + \ldots + X_N$.

Where $X_i = \sqrt{\lambda_i} U_i V_i^T$ are matrices having rank 1; these are called elementary matrices. The collection $(\sqrt{\lambda_i}, U_i, V_i^T)$ will be called the $i$th eigentriple (abbreviated as ET) of the SVD. Vectors $U_i$ are the left singular vectors of the matrix $X$, numbers $\sqrt{\lambda_i}$ are the singular values and provide the singular spectrum of $X$; this gives the name to SSA. Vectors $V_i \sqrt{\lambda_i} = X^T U_i$ are called vectors of principal components (PCs).

3rd step: Eigentriple grouping.

Partition the set of indices $\{1, \ldots, d\}$ into m disjoint subsets $I_1, \ldots, I_m$.

Let $I = \{i_1, \ldots, i_p\}$. Then the resultant matrix $X_I$ corresponding to the group $I$ is defined as $X = X_{I_1} + \ldots + X_{I_m}$. The resultant matrices are computed for the groups and the grouped SVD expansion of $X$ can now be written as $X = X_{I_1} + \ldots + X_{I_m}$.

4th step: Diagonal averaging.

Each matrix $X_{I_j}$ of the grouped decomposition is hankelized and then the obtained Hankel matrix is transformed into a new series of length $N$ using the one-to-one correspondence between Hankel matrices and time series. Diagonal averaging applied to a resultant matrix $X_{I_k}$ produces a reconstructed series $\widetilde{X}^{(k)} = \left( \widetilde{X}_1^{(k)}, \ldots, \widetilde{X}_N^{(k)} \right)$. In this way, the initial series $x_1, \ldots x_N$ is decomposed into a sum of $\boldsymbol{m}$ reconstructed subseries:

$$x_n = \sum_{k=1}^{m} \widetilde{X}_n^{(k)}, (\, n = 1, 2, \ldots, N) \tag{2}$$

This decomposition is the main result of the *SSA* algorithm. The decomposition is meaningful if each reconstructed subseries could be forecasting as a part of either trend or some periodic component or noise.

## 2.2. Forecasting Models and Methods

In this paper, the proposed combined model, based on the nonpositive constraint theory, is integrated with two single statistics forecasting models, two optimization artificial neural network (ANN) forecasting models, and the adaptive network-based fuzzy inference system (ANFIS) based on the Takagi–Sugeno fuzzy inference system.

### 2.2.1. Back-Propagation Neural Network (BPNN) Model

BPNN is a type of multilayer feed-forward neural network with a wide variety of applications. It is based on a gradient descent method that minimizes the sum of the squared errors between the actual and desired output values. The transfer function is of the neuron type. The output function is between 0 and 1 and can transform input to output for continuous nonlinear mapping [32].

The topology of the *BPNN*, is shown as follows:

$$X' = \{X'i\} = 2 \times \frac{X_i - X_{imin}}{X_{imax} - X_{imin}} - 1, (i = 1, 2, \ldots, n), X' \subset [-1, 1] \tag{3}$$

where $X_{min}$ and $X_{max}$ are the minimum and maximum value of the input array or output vectors and $X'i$ denotes the real value of each vector.

Step 1. Calculate outputs of all hidden layer nodes.

$$y_j = f\left( \sum_i w_{ji} x_i + b_j \right) = f(net_j)(i = 1, \ldots, n; j = 1, \ldots, 2n) \tag{4}$$

$$net_j = \sum_i w_{ji} x_i + b_j, (j = 1, \ldots, 2n) \tag{5}$$

where the activation value of node $j$ is $net_j$, $w_{ji}$ represents the connection weight from input node $i$ to hidden node $j$, $b_j$ represents the bias of neuron $j$, $y_j$ represents the output of hidden layer node $j$, and $f$ is the activation function of a node, which is usually a sigmoid function.

Step 2. Calculate the output data of the neural network.

$$O_1 = f_0 \left( \sum_j w_{0j} y_i + b_0 \right), (i = 1, \ldots, 2n) \tag{6}$$

where $w_{0j}$ represents the connection threshold from hidden node $j$ to the output node, $b_0$ represents the bias of the neuron, $O_1$ represents the output data of the network, and $f_0$ is the activation function of the output layer node.

Step 3. Minimize the global error via the training algorithm.

$$Mean\ Square\ Error = \frac{1}{m} \sum (O_1 - Z)^2 \tag{7}$$

where $Z$ represents the real data vector of output, $m$ represents the number of output.

### 2.2.2. Extreme Learning Machine Neural Network Model

The ELMNN is a type of single hidden-layer, feed-forward neural network (SLFN) in which the hidden layer parameters do not need to be tuned [33].

For given dataset $T = \{x_1, t_1, x_2, t_2, \ldots, x_i, t_i\}$ where $x_i = [x_{i1}, x_{i2}, \ldots, x_{in}]^T \in R^m$, $t_i = [t_{i1}, t_{i2}, \ldots, t_{in}]^T \in R^m$ set the activation function which contains $L$ hidden layer nodes as $g(x)$. The computational steps of the standard *ELMNN* are illustrated as follows:

(1) Randomize the bias between the input weights and the hidden layer of the given network as:

$$(a_i, b_i),\ i = 1, 2, \ldots, L \tag{8}$$

(2) The feed forward neural network output of activation function $g(x)$ is expressed as:

$$f_L(x) = \sum_{i=1}^{L} \beta_i G(a_i \times x_i + b_i), a_i \in R^n, \beta \in R^m \tag{9}$$

where the output matrix $H$ is shown as:

$$H(a_1, \ldots, a_L, b_1, \ldots, b_L, x_1, \ldots, x_N) = \begin{bmatrix} G(a_1 \times x_1 + b_1) & \cdots & G(a_L \times x_1 + b_L) \\ \vdots & \ddots & \vdots \\ G(a_1 \times x_1 + b_1) & \cdots & G(a_L \times x_1 + b_L) \end{bmatrix}_{N \times L} \tag{10}$$

Thus formula (10) can be simplified as:

$$H\beta = Y \tag{11}$$

Among it

$$\beta = \begin{bmatrix} \beta_1^T \\ \beta_2^T \\ \vdots \\ \beta_L^T \end{bmatrix}_{L \times m} \tag{12}$$

and

$$Y = \begin{bmatrix} Y_1^T \\ Y_2^T \\ \vdots \\ Y_L^T \end{bmatrix}_{N \times m} \tag{13}$$

(3) Output weight matrix $\beta$ can be obtained by the following formula:

$$\beta = H^+ Y \tag{14}$$

where $H^+$ represents generalized inverse matrix of hidden layer output matrix.

### 2.2.3. Fuzzy Inference Systems Model: Adaptive Neuro-Fuzzy Training of Sugeno-Type FIS

An adaptive neuro-fuzzy inference system or adaptive network-based fuzzy inference system (ANFIS) is a kind of artificial neural network based on the Takagi–Sugeno fuzzy inference system. The technique was developed in the early 1990s. Since it integrates neural networks and fuzzy logic principles, it can potentially capture the benefits of both in a single framework. Its inference system corresponds to a set of fuzzy if–then rules that have learning capability to approximate nonlinear functions, as shown in Figure 1 [34].

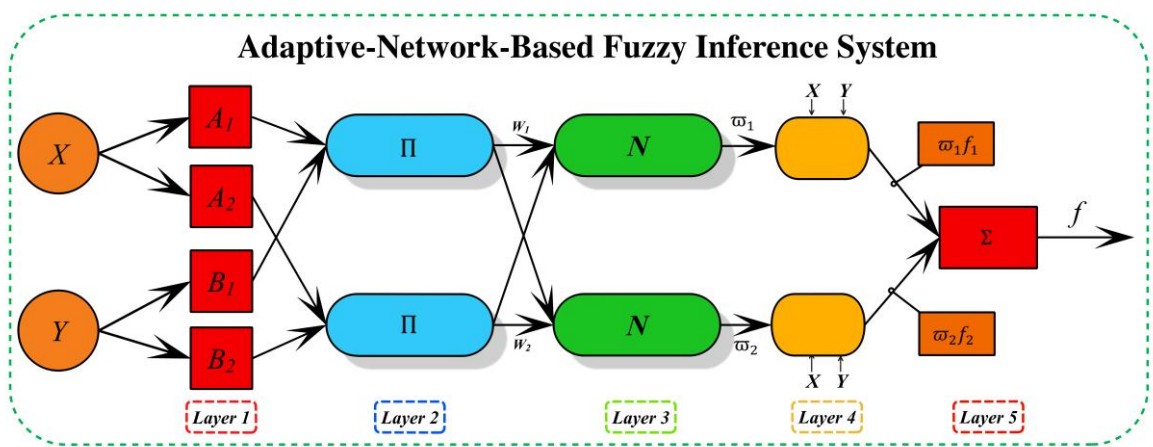

**Figure 1.** The flow of the adaptive network-based fuzzy inference system (ANFIS).

### 2.2.4. Statistical Linear Models: Autoregressive Integer Moving Average and Holt Winters

The autoregressive integer moving average (ARIMA) model is one of the most popular forecasting models [35]. The ARIMA model can be expressed as follows:

$$y_t = \varphi_1 y_{t-1} + \varphi_2 y_{t-2} + \ldots + \varphi_p y_{t-p} + \varepsilon - \theta_q \varepsilon_{t-q} \tag{15}$$

where $y_i (i = 1, 2, \ldots, t)$ is the actual value, $\varepsilon_i (i = 1, 2, \ldots, t)$ is the random error at time $t$, $\varphi_i$ and $\theta_i$ represent the coefficients, and $p$ and $q$ are inter-numbers that are often referred to as autoregressive and moving average polynomials, respectively [36].

The output of Holt–Winters (HW) method is written as $F_{t+m}$, an estimate of the value of $x$ at time $t + m$, $m > 0$ based on the raw data up to time t and suppose we have a sequence of observations $\{x_t\}$, beginning at time $t = 0$ with a cycle of seasonal change of length $L$. The formula and recursive updating equations are following:

$$F_{t+m} = s_t + mb_t + c_{t-L+1+(m-1)} \tag{16}$$

where, $s_t = \alpha(x_t - c_{t-L}) + (1 - \alpha)(s_{t-1} + b_{t-1})$, $b_t = \beta(s_t - s_{t-1}) + (1 - \beta)b_{t-1}$, $c_t = \gamma(x_t - s_{t-1} - b_{t-1}) + (1 - \gamma)c_{t-L}$, $\alpha$ is the data smoothing factor, $\beta$ is the trend smoothing factor, $\gamma$ is the seasonal change

smoothing factor, these three parameters are between 0 to 1. $\{s_t\}$ and $\{b_t\}$ represent the smoothed value of the constant part for time t and the sequence of best estimates of the linear trend that are superimposed on the seasonal changes, respectively. $\{c_t\}$ is the sequence of seasonal correction factors.

### 2.2.5. Combined Model

The combined forecasting theory [29] states that, to solve a certain forecasting problem, which could be solved by M types of forecasting models, the weight coefficients should be properly selected, and then the results from several forecasting methods should be added up. In this condition, it has been regarded as an improvement on single models and an effective and simple way to improve forecasting stability [37].

This paper proposes a new combined model, based on weight-coefficient optimization. In order to improve the forecasting accuracy of the combined model, we modified cuckoo search method with steepest descent to solve the slow speed of convergence and the imprecise accuracy of convergence during the later period of CS optimization. Experiment V shows that the optimizing performance of SDCS is better than that of CS. Thus, SDCS was employed to optimize the weight coefficients of the combined model. The proposed combined model based on nonpositive constraint theory (NNCT) consolidates several models including two hybrid neural networks, a fuzzy model and two linear models to ensure to take advantages of each models.

**Definition 1.** *The traditional forecasting combined method attempts to find the best weight of the combined models based on minimizing SSE:*

$$\min J = L^T EL = \sum_{t=1}^{T} \sum_{j=1}^{m} \sum_{i=1}^{m} l_i l_j e_{it} e_{jt} \begin{cases} R^T L = 1 \\ L \geq 0 \end{cases} \tag{17}$$

*where* $L = (l_1, l_2, \ldots, l_m)$ *is the weight vector,* $R = (1, 1, \ldots, 1)^T$ *is a column vector where all elements are 1, and* $E_{ij} = e_i^T e_j$, *where* $e_i = (e_{i1}, e_{i2}, \ldots, e_{in})$ *is called the error information matrix*

**Definition 2.** *An improved of the traditional combined method based on the nonnegative constraint theory (NNCT) is given as follows:*

$$\min J = L^T EL = \sum_{t=1}^{T} \sum_{j=1}^{m} \sum_{i=1}^{m} l_i l_j e_{it} e_{jt}, s.t \ R^T L = 1 \tag{18}$$

The weights have no limitation in the range [0, 1]. The experimental results show that when the weight vector has a value in the range [–2, 2], the combined model can obtain desirable results, and this method was regarded as the nonpositive constraint theory (NPCT) [30]. This section provides a weight-determined method that was assessed by experimental simulation rather than a theoretical proof.

The branch of combined model proposed in this paper are Adaptive particle swarm optimization ant colony optimization (APSOACO)-BPNN, DE-ELMNN, ANFIS, ARIMA, and HW, which is shown in Section 3.

### 2.3. Heuristic Algorithm

Many parameters, weights, thresholds, and initial values are required for neural networks, and different parameters have a great impact on the output results. Inspired by nature, many methods have been found by natural laws to solve practical problems. The methods that are inspired by the laws and rules of nature are often called heuristic algorithms. Single heuristic algorithms have limited ability to solve problems, and hybrid algorithms that combine the characteristics of different meta-heuristic algorithms have gradually become a research hot point.

2.3.1. Differential Evolution Algorithm

Similar to other evolutionary algorithms, the differential evolution (DE) algorithm is a stochastic model that simulates biological evolution. Differential evolution is an algorithm that optimizes a problem by trying to improve candidate solutions through iteration. Compared to the evolutionary algorithm, the differential evolution algorithm is based on the population global search strategy using coding, based on differences of a simple mutation operation and the competition strategies, reducing the complexity of genetic manipulation. At the same time, the differential evolution algorithm has specific memory so that it can dynamically track the current search condition to adjust the searching strategy, and has a strong global convergence ability and robustness [38].

2.3.2. Hybrid Adaptive Particle Swarm Optimization based on Ant Colony Optimization

Adaptive particle swarm optimization (APSO) is presented in this paper to improve the accuracy of search capability [39]. As the value of the particle fitness function changes, the inertia weight will be automatically adjusted to enhance the illumination direction of the particle search. Although APSO does not converge fast, it does not fall into local extreme points lightly. Ant colony optimization (ACO), inspired by the action of ant colonies searching for food [40,41], is used for continuous search space optimization problems and is tested on some benchmark functions as well [42].

APSOACO was developed inspired by APSO. The advantages of ant ACO, foraging behavior, and velocity update of adaptive particle swarm optimization APSO were assembled in APSOACO. A sigmoid function is used to convert distance and velocity into heuristic values. The advantages of the hybrid algorithm are as follows: (1) it avoids convergence to a local optimum, (2) it provides a better solution within fewer iterations, i.e., fast convergence, and (3) it has low computational complexity. The basic steps of APSOACO are shown in Algorithm 1 and Figure 2.

---

**Algorithm 1: APSOACO**

---

  *Input:*
  $x_p^{(0)} = \left( x_1^{(0)}, x_2^{(0)}, \ldots, x_q^{(0)} \right)$—a sequence of training data.
  $x_q^{(0)} = \left( x_{q+1}^{(0)}, x_{q+2}^{(0)}, \ldots, x_{q+d}^{(0)} \right)$—a sequence of verifying data.
  *Output:*
  $\alpha_{best}$—the value of with the best fitness value in particle searching space.
  *Parameters:*
 *APSO:*
  *Particles* = 30, $c_1 = c_2 = 2$, $w_0 = 1$, *Maximum iteration* = 300
 *Stopping criteria* = maximum iteration
 *ACO:*
  *NC_max*—Maximum iterations:50
  *m*—The number of ant:30
  *Alpha*—Parameters of the important degree of information elements:1
  *Beta*—Parameters of the important degree of the Heuristic factor:5
  *Rho*—Parameters of the important degree of the heuristic factor:0.1
  *Q*—Pheromone increasing intensity coefficient:100

---

1: /*Initialize *popsize* candidates with the values between 0 and 1*/

2: **FOR EACH** $i\ 1\ \le\ i\ \le\ n$ **DO**

3: $\alpha_i^1 =\ rand(m,n)$

4: **END FOR**

5: $P\ =\ \left\{\alpha_i^{iter} : 1\ \le\ i\ \le\ popsize\right\}$

6: *iter* = 1; Evaluate the corresponding fitness function $F_i$

7: /* Find the best value of repeatedly until the maximum iterations are reached. */

8: **WHILE** ($gbest > pbest_i$) **DO**

9: /* Find the best fitness value for each candidates */

10: **FOR EACH** $\alpha_i^{iter}\ \in\ P$ **DO**

11: Build *neural network* by using $x_s^{(0)}$ with the $\alpha_i^{iter}$ value

12: Calculate $x_p^{(0)} = (x_{p+1}^{(0)}, x_{p+2}^{(0)}, \ldots, x_{p+d}^{(0)})$ by *neural network*

13: /*Choose the best fitness value of the $i^{th}$ candidate in history */

14: **IF** ($pBest_i > fitness(\alpha_i^{iter})$) **THEN**

15: $pBest_i = fitness(\alpha_i^{iter})$

16: **END IF**

17: **END FOR**

18: /* Choose the candidate with the best fitness value of all the candidates */

19: **FOR EACH** $\alpha_i^{iter} \in P$ **DO**

20: **IF** ($gBest > pBest_i$) **THEN**

21: $gBest = pBest_i = x_{t+1}^k = x^{gbest}$, $t = 1, 2, \ldots, T$

22: $\alpha_{best} = \alpha_i^{iter}$

23: **END IF**

24: **END FOR**

25: /*Update the values of all the candidates by using ACO's evolution equations. */

26: **FOR EACH** $\alpha_i^{iter}\ \in\ P$ **DO**

27: $\alpha_{t+1} = 0.1 \times \alpha_t$

28: $x^{gbest} = x^{gbest} + (x^{gbest} \times 0.01) \rightarrow \begin{cases} \text{if } f\left(x^{gbest}\right) - f\left(x^{gbest}\right) \ge 0 \rightarrow \text{the sign is } (+) \\ \text{if } f\left(x^{gbest}\right) - f\left(x^{gbest}\right) < 0 \rightarrow \text{the sign is } (-) \end{cases}$

29: **END FOR**

30: $P = \left\{\alpha_i^{iter} : 1\ \le\ i\ \le\ popsize\right\}$

31: *iter* = *iter*+1

32: **END WHILE**

33: **IF NOT**

34: /*Compute solution for each particle by the Adaptive particle swarm optimization */

35: **FOR EACH** $j\ 1\ \le\ j\ \le\ n$, $n = 1, 2, \ldots, 300$ ($n = $ *Maximum iteration* = 300) **DO**

36: $c_1 = c_2 = 2$

37: $w_0 = 1$

38: /*Find the fitness solution *pbest* and *gbest* until the maximum iterations */

39: **WHILE**

40: $V_i\ (t+1) =\ w(t)v_i(t) + c_1 r_1[pbest_i - x_i(t)] + c_2 r_2[pbest - x_i(t)]$

41: $x_i(t+1) =\ x_i(t) + v_i(t+1)$

42: **END**

43: /*Update the values of all the candidates by using APSO's evolution equations. */

44: **FOR EACH** $w(t+1) = \begin{cases} \lambda w(t) + \theta \frac{f(gbest) - f(x_i(t))}{f(gbest) - f(x_{min}(t))} & w(t) > w_{min} \\ w_{min} & otherwise \end{cases}$ **DO**

45: **END**

46: **RETURN** $\alpha_{best}$

### 2.3.3. Hybrid Cuckoo Search Method Based on Steepest Descent

The cuckoo search (CS) algorithm was derived from the action of cuckoos laying their eggs in other birds' nests to let those birds hatch the eggs for them [43]. However, once the host birds discover the cuckoo eggs, they will throw away the eggs or abandon their nests and build a new nest elsewhere. In CS, every nest stands for a solution. The CS algorithm was constructed based on three assumptions: (a) only one egg is laid by each cuckoo in a selected nest randomly; (b) succeeding generations would begin in the best nest; and (c) it is a constant of the number of available host nests; the probability value of the host bird discovering the egg laid by a cuckoo is p, which has a range of 0 to 1.

Similar to other meta-heuristic algorithms, the original CS algorithm is simple and efficient; however, it has disadvantages, such as insufficient search vigor and slow search speed during the latter part of the search. Therefore, this paper proposes an improved CS algorithm, which we call the SDCS model, based on the steepest descent (SD) method [44].

As one of the oldest optimization algorithms, the steepest descent method is simple and intuitive. Currently, many effective optimization algorithms have been established on the basis of this algorithm. In the cause of avoiding slow convergence rate of CS's shortcoming of slow convergence rate, the steepest descent method is used to modify the cuckoo search method, and the modified process can be expressed by the following steps:

Step 1. Select the initial points $x^0$, and give the end error $\varepsilon > 0$. Make $k = 0$.
Step 2. Calculate $\nabla f(x^k)$. If

$$\|\nabla f(x^k)\| = \|-\nabla g(x_i^k)\| < \alpha \bigoplus Levy(\lambda) \leq \varepsilon;\ Levy(\lambda) = t^{-\lambda} \tag{19}$$

stop iterations and output $x^k$. where, $\nabla$ is gradient operator, $\alpha > 0$ is the step size related to the scales of the problem of interest, while the product $\bigoplus$ means entry-wise multiplication, $Levy(\lambda)$ is a Levy flight, which represents the most powerful features of the cuckoo search to generate new eggs provided by a random walk and t is the iteration number. Otherwise go to Step 3.

Step 3. Take

$$p^k = -\nabla f(x^k) \tag{20}$$

Step 4. Conduct one-dimensional search. Solve $t_k$, make

$$f(x^k + t_k p^k) = \min_{t \geq 0} f(x^k + t p^k) \tag{21}$$

Make $x^{k+1} = x^k + t_k p$, $k = k + 1$, go to Step 2.

The step size and step-length distribution function of the cuckoo search algorithm can be improved by using steepest descent due to its simplicity and flexibility. The final optimal solution can be obtained by modifying the step size and step-length distribution function constantly.

### 2.4. Hybrid Models

Section 2.2 introduced the different forecasting models and Section 2.3 introduced some heuristic algorithms. In this section, APSO, ACO, and APSOACO are optimized to the weights and biases of the nonlinear back propagation neural network as shown Figure 2, utilizing the differential evolution (DE) algorithm to optimize the weights and biases of the nonlinear ELMNN (shown in Figure 2) and using a modified cuckoo search algorithm based on steepest descent (SDCS) to optimize the weight vector of the combined forecasting model, which will provide the minimum error for each model (shown in Figure 3).

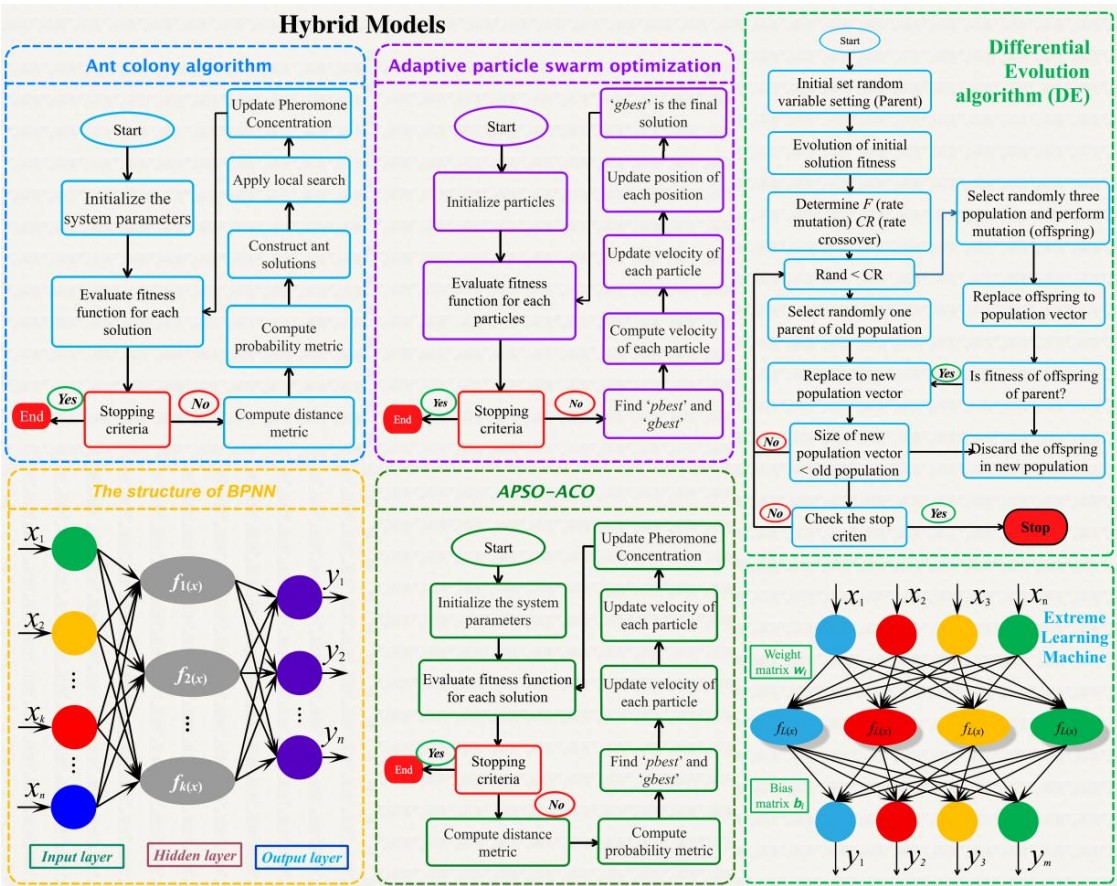

**Figure 2.** The flow of three different hybrid back-propagation neural networks (BPNN) and a hybrid extreme learning machine (ELM) neural network.

## 3. Framework of the Proposed Forecasting Models

In this paper, five forecasting techniques (APSOACO-BPNN, DE-ELMNN, ANFIS, ARIMA, and HW) were used, due to their forecasting ability to solve nonlinear and linear problems. When the data are linear, the linear models ARIMA and HW have good forecasting ability. When the data are nonlinear, such as wind speed data, which is often random and intermittent, the performance of these models is not very good. Therefore, this study presents a combined linear and nonlinear model to solve the forecasting task. The structure of the proposed combined model, which has three phases, is shown in Figure 3.

Phase I: Utilize SSA to preprocess the original short-term wind speed series. In this phase, according to the observed time series, the trajectory matrix is constructed with the wind speed data, and the matrix is decomposed and reconstructed, and different composition on behalf of the original time series signal is extracted, thus the structure of time series analysis, and can further improve the prediction precision.

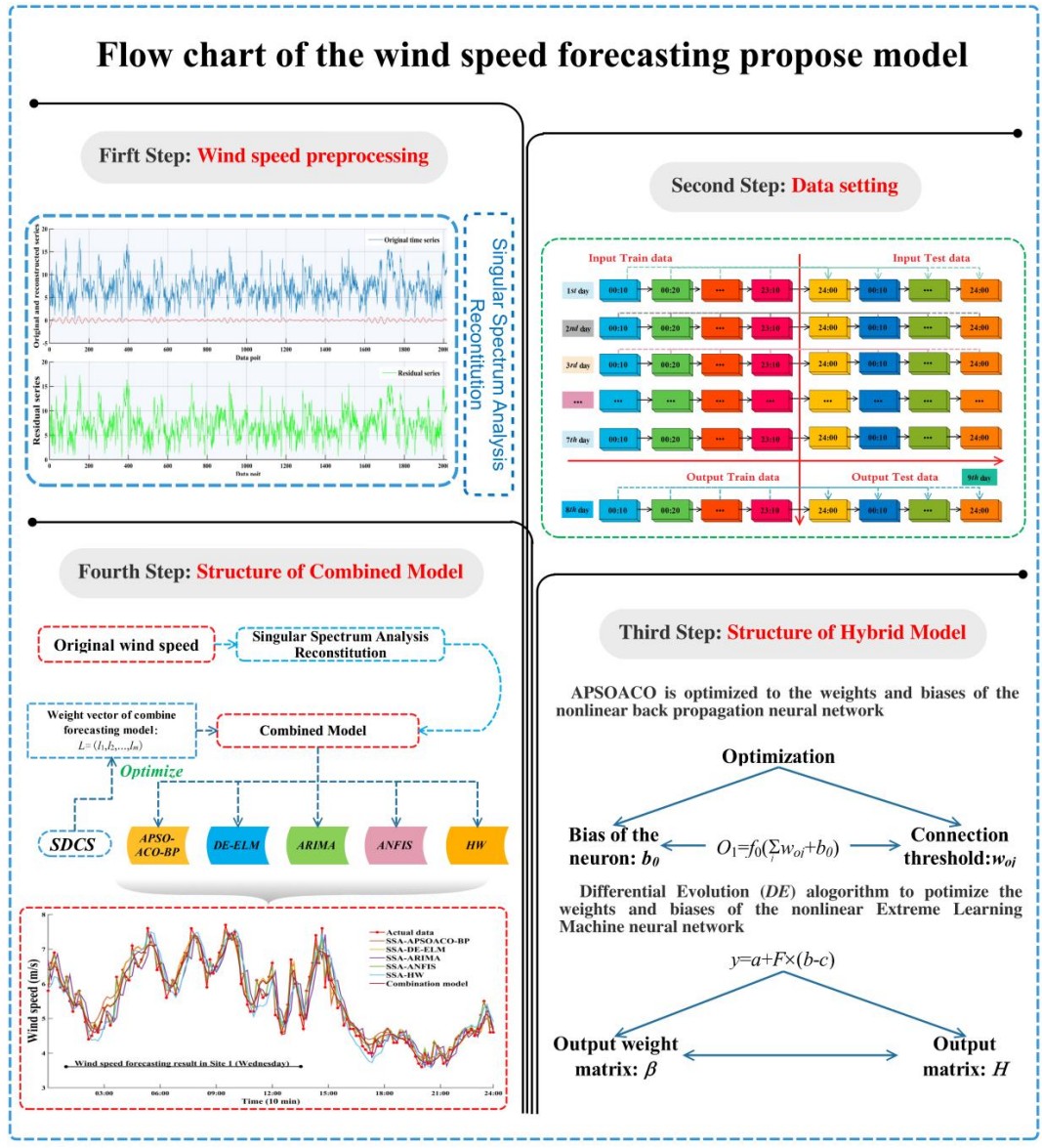

**Figure 3.** The flow of proposed combined model.

Phase II: Construct the test wind speed datasets, forecasting time series wind speed data with different forecasting models. It is worth pointing out that the original wind speed time series is divided into training sets and test sets. In the process of training the network, the input is the filtered data and the output is the original time series of the training set. In the test step, the input is also the filtered data, and the output is the original data.

Phase III: Utilize the test data to choose the branch models of combined model. For different models, every model performance different in the test data. And we also used optimization algorithms to optimize BPNN and ELMNN, and we chose the best hybrid models to build the combined model.

Phase IV: Combine the single forecasting models by the combined model theory. Combine the independent forecasts generated by the aforementioned forecasting engines. NPCT is utilized to predict the distribution of future wind speed with the independent forecasts generated by the forecasting engines as input in different forecasting horizons. NPCT is used to predict the distribution of time series wind speed data, and the independent prediction generated by the prediction engine is used as input to different prediction fields. Hereafter, the combined model can be used to forecast the time series wind speed data.

## 4. Experimental Study

In order to verify the effectiveness of our proposed combined model, 10-min wind speed series were applied to test the performance of these models. Using the wind speed data, which included samples from every wind speed observation site at two wind farms in China in 2015, the effectiveness and reliability of the combined model are verified. In our experiments, the size of training datasets is limited by the moving window method and the size is set as 2016 samples for Site 1 in March, Site 2 in June, Site 3 in September, and Site 4 in November.

There are four experiments in this paper.

Experiment I aims to research the feature of time series wind speed data. In the process of wind speed forecasting, it is a challenging task to choose which forecasting models are employed to forecast the wind speed. Experiment I uses linear and nonlinear functions to test the feature of the time series wind speed data.

The developed hybrid BPNN with other two hybrid forecasting models, APSO-BPNN and ACO-BPNN, are compared in Experiment II. We also compared single optimization and hybrid optimization, optimizing the weight and threshold of the nonlinear BPNN. In Experiment II, it also shown the differential evolution optimizes the weight matrix of hidden and output layers and the bias matrix between the two layers to compare the single ELMNN and DE-ELMNN

Experiment III shows the performance of each forecasting model at each time point (hours). We compared two single linear forecasting models, a single fuzzy forecasting model and two hybrid nonlinear models with the forecasting results of Monday from Site 1.

Experiment VI presents the forecasting results of branch models and combined model from four sites to test the performance of models in different sites.

Experiment V shows the test between SDCS and CS, which reveals SDCS is better than CS.

### 4.1. Metric of Forecasting Model and Experiment Setup

Accuracy of results is a significant criterion in this paper. Four metrics, average error (AE), mean absolute error (MAE), mean square error (MSE), and mean absolute percentage (MAPE), as shown in Table 1, are used to judge the performance of the forecasting models.

**Table 1.** Four metric rules. AE: average error; MAE: mean absolute error; MSE: mean square error; MAPE: mean absolute percentage error.

| Metric | Definition | Equation |
|--------|-----------|----------|
| AE | The average forecast error of $n$ times forecast results | $AE = \frac{1}{N} \sum_{n=1}^{N} (y_n - \hat{y}_n)$ |
| MAE | The average absolute forecast error of $n$ times forecast results | $MAE = \frac{1}{N} \sum_{n=1}^{N} |y_n - \hat{y}_n|$ |
| MSE | The average of the prediction error squares | $MSE = \frac{1}{N} \sum_{n=1}^{N} (y_n - \hat{y}_n)^2$ |
| MAPE | The average of absolute percentage error | $MAPE = \frac{1}{N} \sum_{n=1}^{N} \left| \frac{y_n - \hat{y}_n}{y_n} \right| \times 100\%$ |

### 4.2. Experiment I: Use the Linear and Nonlinear Functions to Test the Feature of Wind Speed Series

For the data, only with a better understanding of the features of the data can we better select the model to prepare for future work. In order to achieve better results, we must consider the characteristics of the data. Generally speaking, the linear model has a better fitting effect for linear data, as the nonlinear model does for nonlinear data.

Only when we understand the characteristics of the data can we achieve good results in future forecasting work. For the data, it is not just linear or nonlinear, but both. Therefore, it is necessary to judge the linear nonlinearity of the data used in this paper, so we constructed the following experiments.

From the results of Table 2, wind speed data are both linear and nonlinear by hypothesis test. So, the linear models and nonlinear models considered in our proposed forecasting model are correct and necessary.

**Table 2.** (**A**) Testing wind speed data by adjusting to linear functions or nonlinear functions. (**B**) The explanations of the test parameters.

| | | Number of Observations | Error Degrees of Freedom | *R*-Squared | Adjusted *R*-Squared | F-statistic vs. Constant Model | *p*-Value |
|---|---|---|---|---|---|---|---|
| | | | | **(A)** | | | |
| $y \sim f_1(x)$ | Site 1 | 1003 | 997 | 0.941 | 0.941 | 3180 | 0 |
| | Site 2 | 1003 | 997 | 0.936 | 0.935 | 2900 | 0 |
| | Site 3 | 1003 | 997 | 0.933 | 0.932 | 2760 | 0 |
| | Site 4 | 1003 | 997 | 0.905 | 0.904 | 1890 | 0 |
| $y \sim f_2(x)$ | Site 1 | 1003 | 997 | 0.932 | 0.932 | 9020 | 0 |
| | Site 2 | 1003 | 997 | 0.931 | 0.931 | 9700 | 0 |
| | Site 3 | 1003 | 997 | 0.927 | 0.927 | 10500 | 0 |
| | Site 4 | 1003 | 997 | 0.881 | 0.881 | 6990 | 0 |
| $y \sim f_3(x)$ | Site 1 | 1003 | 997 | 0.919 | 0.918 | 7490 | 0 |
| | Site 2 | 1003 | 997 | 0.918 | 0.918 | 8140 | 0 |
| | Site 3 | 1003 | 997 | 0.910 | 0.910 | 8430 | 0 |
| | Site 4 | 1003 | 997 | 0.880 | 0.880 | 6940 | 0 |
| | | | | **(B)** | | | |
| Number of Observations | | Number of Rows without Any Not a Number (NaN) Values. | | | | | |
| Error Degrees of Freedom | | $n - p$, where $n$ is the number of observations, and $p$ is the number of coefficients in the model, including the intercept. | | | | | |
| R-squared and Adjusted R-squared | | Coefficient of determination and adjusted coefficient of determination, respectively | | | | | |
| F-statistic vs. Constant Model | | Test statistic for the *F*-test on the regression model. It tests for a significant regression relationship between the response variable and the predictor variables | | | | | |
| *p*-Value | | *p*-value for the *F* statistic of the hypotheses test that the corresponding coefficient is equal to zero or not. | | | | | |

**Note**: In order to verify the linear or nonlinear character of wind speed, three functions were structured: (1) linear function $f_1(x) = 1 + \sum_{i=1}^{5} b_i x_i$; (2) nonlinear function $f_2(x) = 1/(1 + exp(\sum_{i=1}^{5} b_i x_i) + b_6)$; and (3) nonlinear function $f_3(x) = 2/(1 + exp(\sum_{i=1}^{5} b_i x_i + b_6)) - 1$.

### 4.3. Experiment II: Forecasting Results of Hybrid Models and Single Models

In Section 2.4, we proposed four hybrid models, including ACO-BPNN, APSO-BPNN, APSOACO-BPNN, and DE-ELMNN. In this section, the performance of three different hybrid BPNN are compared to certify the hybrid BPNN of combined can obtain better results compared to other hybrid BPNNs. At the same time, ELMNN and DE-ELMNN are also compared in order to select a better model.

#### 4.3.1. Comparesion of Two ELMNNs: ELMNN and DE-ELMNN

A novel and fast learning neural network, ELMNN, is based on modification of the traditional single hidden-layer feed-forward. The obvious advantage is that it randomly assigns the weights and thresholds of the input layer and the hidden layer in the learning process, which is not required to accommodate to these parameters in the learning process, so that the training process is extraordinarily fast. However, the random numbers initialize the input weights and hidden biases. The learning process of ELMNN has poor stability, the same as the other neural network. Then the output weights are calculated through an inverse operation on the hidden layer output matrix, which is randomly

determined according to Equation (10). Section 2.3 of this paper introduces how to use the DE algorithm to search the weights and threshold values for ELMNN. In this experiment the performance of hybrid model and single model, SSA-DE-ELMNN and SSA-ELMNN, were tested.

Table 3 show the forecasting results of Sites 1–4 in a week, it is clear that the DE-ELMNN model performs much better than the other three models. To explain the results of the proposed method, we utilize the first site as an example. First, the hybrid DE-ELMNN model has the smallest statistical error of AE, MAE, MSE, and MAPE when compared with the single ELMNN. As Table 3 shown, for an example, from the Monday results in site 1, the MAPE of our hybrid DE-ELMNN is 5.85% while the single ELMNN is calculated to be 6.30%; thus, the precision is improved by 0.45% and the most accuracy improved is from Wednesday in site 1 with the value 0.98%. The DE-ELMNN improved MAPE value 0.01% at least from Table 3. Second, in this condition, the results reveal the DE-ELMNN is more effective than DE-ELMNN.

**Table 3.** The forecasting results of the two different ELM neural network in Site 1–4. SSA: singular spectrum analysis; DE: differential evolution.

| Sites | Model | Index | Mon | Tue | Wed | Thu | Fri | Sat | Sun |
|---|---|---|---|---|---|---|---|---|---|
| Site 1 | SSA-ELM | AE | −0.0777 | −0.0005 | 0.0100 | −0.0174 | −0.0094 | 0.0176 | −0.0343 |
| | | MAE | 0.3848 | 0.4054 | 0.2589 | 0.3734 | 0.3052 | 0.2521 | 0.2458 |
| | | MSE | 0.2957 | 0.2868 | 0.1232 | 0.2289 | 0.1457 | 0.1022 | 0.0940 |
| | | MAPE | 6.30% | 3.76% | 4.74% | 5.14% | 5.83% | 4.11% | 3.26% |
| | SSA-DE-ELM | AE | −0.0617 | 0.0697 | −0.0005 | 0.0526 | −0.0189 | 0.0383 | −0.0269 |
| | | MAE | 0.3425 | 0.3832 | 0.4054 | 0.2637 | 0.3020 | 0.2472 | 0.2339 |
| | | MSE | 0.1935 | 0.2523 | 0.2868 | 0.1065 | 0.1442 | 0.1007 | 0.0897 |
| | | MAPE | 5.85% | 3.58% | 3.76% | 4.87% | 5.77% | 4.05% | 3.12% |
| Site 2 | SSA-ELM | AE | −0.0358 | 0.0357 | −0.0071 | −0.1277 | −0.0161 | 0.0255 | −0.0563 |
| | | MAE | 0.3665 | 0.4391 | 0.2621 | 0.5243 | 0.2792 | 0.2786 | 0.2535 |
| | | MSE | 0.2062 | 0.3079 | 0.1034 | 0.5319 | 0.1239 | 0.1254 | 0.1198 |
| | | MAPE | 5.31% | 3.80% | 4.05% | 5.98% | 4.92% | 3.66% | 2.97% |
| | SSA-DE-ELM | AE | 0.0977 | −0.0423 | 0.0574 | −0.0784 | 0.0521 | −0.0037 | −0.0265 |
| | | MAE | 0.3599 | 0.3816 | 0.3449 | 0.4945 | 0.3021 | 0.2344 | 0.2567 |
| | | MSE | 0.2042 | 0.2343 | 0.1964 | 0.4236 | 0.1592 | 0.1012 | 0.1252 |
| | | MAPE | 5.16% | 3.43% | 5.17% | 5.42% | 5.20% | 3.35% | 2.98% |
| Site 3 | SSA-ELM | AE | −0.0213 | 0.0493 | 0.04047 | 0.0164 | −0.0199 | 0.0079 | −0.0103 |
| | | MAE | 0.3511 | 0.5322 | 0.26225 | 0.3971 | 0.3107 | 0.3094 | 0.2737 |
| | | MSE | 0.2061 | 0.4208 | 0.10542 | 0.2831 | 0.1613 | 0.1785 | 0.1231 |
| | | MAPE | 5.23% | 4.59% | 4.13% | 4.24% | 5.30% | 4.39% | 3.24% |
| | SSA-DE-ELM | AE | 0.065 | 0.0222 | −0.0144 | 0.0028 | −0.0185 | 0.0768 | −0.0165 |
| | | MAE | 0.3055 | 0.5092 | 0.2513 | 0.3870 | 0.2989 | 0.3016 | 0.2683 |
| | | MSE | 0.1726 | 0.3872 | 0.0943 | 0.2507 | 0.1552 | 0.1606 | 0.1204 |
| | | MAPE | 4.48% | 4.41% | 3.92% | 4.15% | 5.12% | 4.26% | 3.16% |
| Site 4 | SSA-ELM | AE | −0.0233 | 0.0389 | 0.0053 | −0.0333 | −0.0011 | 0.0205 | −0.0177 |
| | | MAE | 0.3005 | 0.5811 | 0.3285 | 0.4540 | 0.3047 | 0.2694 | 0.3213 |
| | | MSE | 0.1456 | 0.6113 | 0.1459 | 0.3896 | 0.1747 | 0.1111 | 0.1706 |
| | | MAPE | 5.13% | 5.78% | 4.43% | 5.56% | 5.63% | 3.72% | 4.22% |
| | SSA-DE-ELM | AE | 0.0117 | −0.0424 | 0.0128 | 0.0222 | −0.015 | 0.0148 | 0.0414 |
| | | MAE | 0.2958 | 0.5658 | 0.2963 | 0.2971 | 0.2997 | 0.2996 | 0.3071 |
| | | MSE | 0.1384 | 0.5411 | 0.1237 | 0.3243 | 0.1623 | 0.1056 | 0.1565 |
| | | MAPE | 5.01% | 5.62% | 4.27% | 5.26% | 5.45% | 3.62% | 3.93% |

Table 3 and Figure 4A show that the MAPE, MAE, and MSE values of DE-ELMNN are 3.58%, 0.3832 and 0.2523, respectively in Tuesday from site 1, which are lower than ELMNN. And for other days from site 1, it is clearly shows DE-ELMNN performance better than ELMNN. From the forecasting errors shown in Figure 4B, the error of DE-ELMNN is approximately 0 than ELMNN especially in Wednesday. Figure 4 also shows the 95% confidence intervals (CIs) obtained by DE-ELMNN and ELMNN, the figure indicates that both the upper and lower CI are close between DE-ELMNN and ELMNN, but for DE-ELMNN there are more points in the confidence interval.

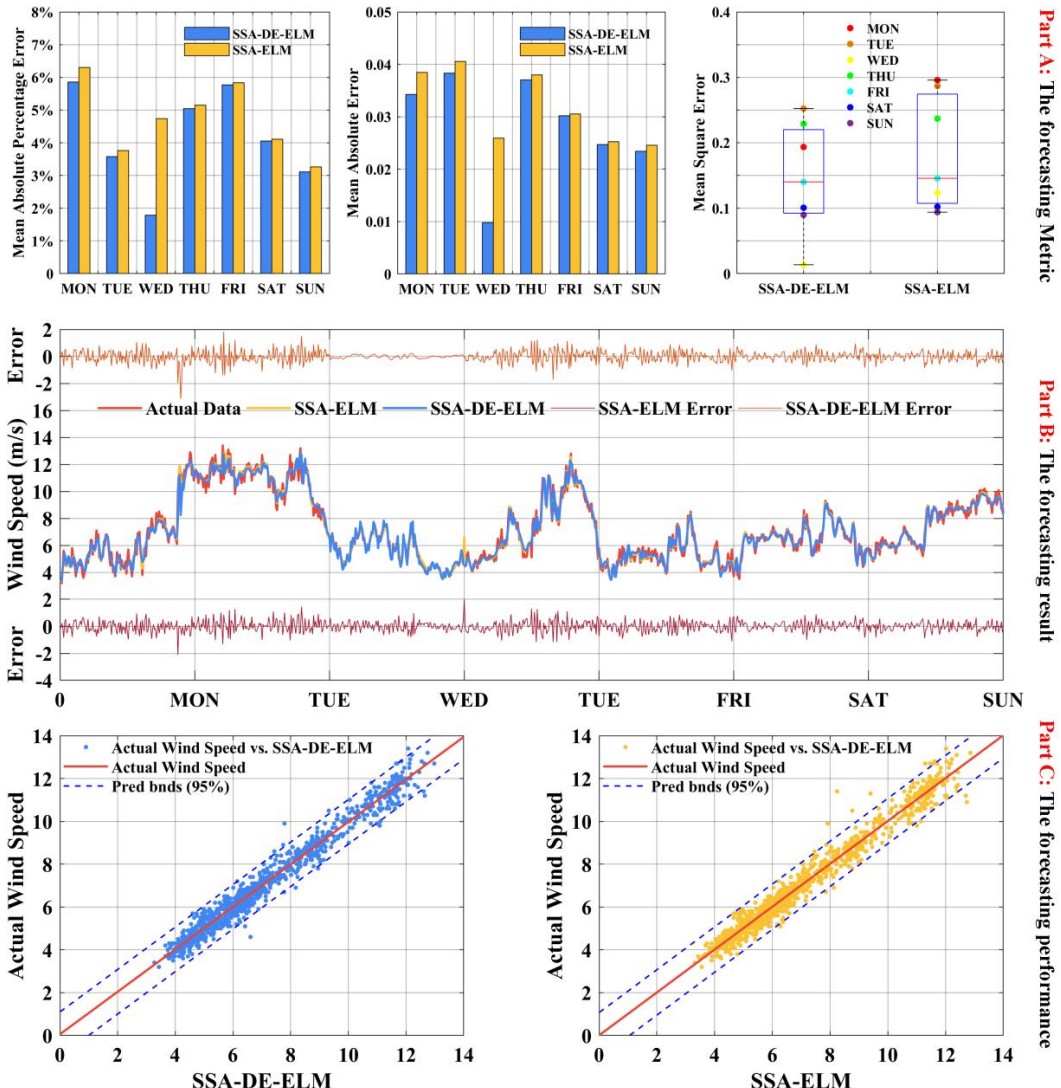

**Figure 4.** The results of two different ELMNN forecasting models in site 1.

**Remark:** *From the Table 3 and Figure 4, the results indicate that SSA-DE-ELMNN shows better performance than the single SSA-ELMNN. In brief, it can be explained that SSA, which is able to denoise the time series wind speed data as a preprocessing method, and the DE algorithm, which has better ability to detect optimizing parameters for BPNN, can improve forecasting accuracy. Furthermore, due to the vertical dataset selection method, the structure of the dataset is optimal, DE-ELMNN can reach the minimum error effectively.*

4.3.2. Comparesion of Three Optimization BPNNs: ACO-BPNN, APSO-BPNN, and APSOACO-BPNN

The traditional BPNN was comprehensively applied to wind speed forecasting, but as it has defects, it is easy to fall into local minimum and low forecasting accuracy. So Section 2.3 introduces the modified ant colony optimization as a way to use APSOACO to optimize the BPNN as a strong predictor to improve the accuracy of forecasting results.

This experiment tested the three optimization nonlinear BPNNs to choose the best performance of hybrid nonlinear BPNN as a portion of the combined forecasting model. APSO and ACO have many advantages. ACO has strong global search ability, is robust, and is easy to combine with other algorithms, and APSO has fast convergence speed and fewer parameters, and is simple and easy to operate. With ACO the slow convergence rate occurs easily, but at the same time APSO easily falls into local extreme points. All of these defects will affect the parameters with which the algorithms

are optimized. Combining the advantages and disadvantages of ACO and APSO, the APSOACO algorithm is proposed to ameliorate the capacity of parameter optimization.

It is clear that, as the results show in Table 4, the hybrid SSA-APSOACO-BPNN model is superior to SSA-APSO-BPNN and SSA-ACO-BPNN according to the values of MAE, MSE, and MAPE. In addition, the MAPE of the SSA-APSOACO-BPNN is 6.08%, 3.67%, 4.62%, 4.90%, 5.55%, 4.21%, and 3.21% from Monday to Sunday in site 1, respectively. What we could find is that SSA-APSOACO-BPNN not only obtained high accuracy and stability, but also improved the time series wind speed data suited to be forecasted by this model. From the experiments, we can determine the MAE, MSE, AE, and MAPE and whether the proposed SSA-APSOACO-BPNN achieves the best performance in most of tome. It can be seen that, SSA-APSO-BPNN achieved better performance in Sunday of site 3 and SSA-ACO-BPNN obtain better results in Tuesday of site 2 and Saturday of site 3.

Figure 5A show that the MAPE, MAE, and MSE values of APSOACO-BPNN are 0.4116, 0.2663 and 3.57%, respectively in Tuesday from site 2, which are lower than APSO-BPNN and ACO-BPNN. And for other days from site 2, it is clearly shows APSOACO-BPNN performance better than other hybrid BPNNs. Figure 5C also shows the 95% confidence intervals (CIs) obtained by three hybrid BPNNs, the figure indicates that both the upper and lower CI are close between this three hybrid models, but for APSOACO-BPNN there are more points in the confidence interval.

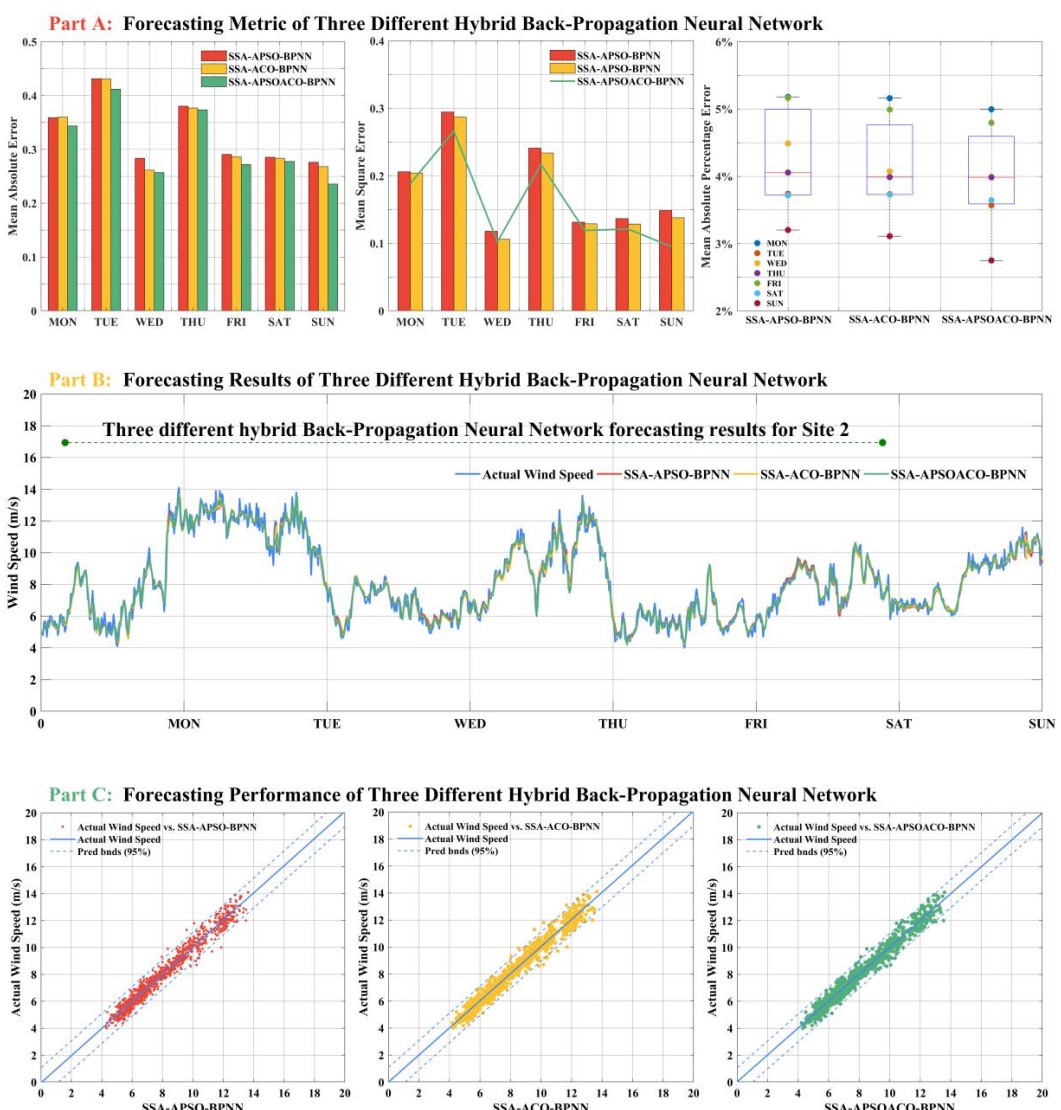

**Figure 5.** The results of three different BPNN forecasting models in site 1.

**Table 4.** The results of the three different *BPNN* forecasting models in Site 1–4.

| Sites | | SSA-APSO-BPNN | | | | SSA-ACO-BPNN | | | | SSA-APSOACO-BPNN | | | |
|---|---|---|---|---|---|---|---|---|---|---|---|---|---|
| | | AE | MAE | MSE | MAPE | AE | MAE | MSE | MAPE | AE | MAE | MSE | MAPE |
| Site 1 | Mon | 0.0013 | 0.3821 | 0.2399 | 6.56% | −0.0805 | 0.3717 | 0.2224 | 6.22% | −0.0716 | 0.3612 | 0.2141 | 6.08% |
| | Tue | 0.0173 | 0.4173 | 0.3150 | 3.86% | 0.0376 | 0.4070 | 0.2835 | 3.80% | 0.0118 | 0.3955 | 0.2763 | 3.67% |
| | Wed | 0.0115 | 0.2602 | 0.1028 | 4.76% | 0.0299 | 0.2546 | 0.0978 | 4.65% | −0.0059 | 0.2421 | 0.0938 | 4.62% |
| | Thu | 0.0061 | 0.3833 | 0.2393 | 5.22% | 0.0235 | 0.3806 | 0.2364 | 5.16% | 0.0050 | 0.3613 | 0.2186 | 4.90% |
| | Fri | 0.0483 | 0.3167 | 0.1547 | 6.15% | 0.0219 | 0.3039 | 0.1438 | 5.87% | −0.0151 | 0.2914 | 0.1356 | 5.55% |
| | Sat | 0.0792 | 0.2696 | 0.1178 | 4.46% | 0.0692 | 0.2728 | 0.1201 | 4.50% | 0.0164 | 0.2565 | 0.1079 | 4.21% |
| | Sun | −0.1141 | 0.2695 | 0.1120 | 3.48% | −0.0657 | 0.2491 | 0.0973 | 3.27% | −0.0564 | 0.2432 | 0.0939 | 3.21% |
| Site 2 | Mon | 0.9634 | 0.3587 | 0.2061 | 5.18% | −0.0564 | 0.3599 | 0.2042 | 5.16% | 0.9643 | 0.3434 | 0.1886 | 5.00% |
| | Tue | 0.8205 | 0.4311 | 0.2945 | 3.74% | 0.0050 | 0.4307 | 0.2872 | 3.74% | 0.8253 | 0.4116 | 0.2663 | 3.57% |
| | Wed | 0.8867 | 0.2834 | 0.1179 | 4.49% | −0.0175 | 0.2616 | 0.1064 | 4.07% | 0.8888 | 0.2570 | 0.1023 | 3.99% |
| | Thu | 0.9326 | 0.3804 | 0.2411 | 4.06% | 0.9347 | 0.3770 | 0.2339 | 3.99% | −0.0502 | 0.3737 | 0.2171 | 3.98% |
| | Fri | 0.8571 | 0.2908 | 0.1313 | 5.16% | 0.0077 | 0.2860 | 0.1289 | 4.99% | 0.8571 | 0.2721 | 0.1192 | 4.79% |
| | Sat | 0.9124 | 0.2854 | 0.1369 | 3.72% | 0.9202 | 0.2837 | 0.1286 | 3.73% | 0.0125 | 0.2774 | 0.1212 | 3.65% |
| | Sun | 0.9443 | 0.2760 | 0.1486 | 3.20% | 0.9474 | 0.2676 | 0.1378 | 3.11% | −0.0032 | 0.2357 | 0.0952 | 2.75% |
| Site 3 | Mon | 0.9701 | 0.3429 | 0.2156 | 4.90% | 0.9645 | 0.3721 | 0.2649 | 5.23% | −0.1005 | 0.3367 | 0.2094 | 4.84% |
| | Tue | 0.7411 | 0.5331 | 0.4305 | 4.59% | 0.0963 | 0.5559 | 0.4612 | 4.81% | 0.7546 | 0.5235 | 0.4094 | 4.52% |
| | Wed | 0.8855 | 0.2575 | 0.0995 | 4.00% | 0.8828 | 0.2571 | 0.0992 | 3.99% | 0.0014 | 0.2539 | 0.0964 | 3.96% |
| | Thu | 0.9222 | 0.3923 | 0.2694 | 4.19% | 0.0337 | 0.3980 | 0.2658 | 4.28% | 0.9218 | 0.3839 | 0.2633 | 4.07% |
| | Fri | 0.8213 | 0.3041 | 0.1595 | 5.22% | 0.0167 | 0.3212 | 0.1757 | 5.45% | 0.8143 | 0.3007 | 0.1629 | 5.14% |
| | Sat | 0.8782 | 0.3228 | 0.1929 | 4.52% | 0.0581 | 0.3031 | 0.1744 | 4.49% | 0.8858 | 0.3091 | 0.1807 | 4.44% |
| | Sun | 0.9528 | 0.2751 | 0.1218 | 3.28% | 0.0418 | 0.2787 | 0.1236 | 3.33% | 0.9516 | 0.2787 | 0.1237 | 3.33% |
| Site 4 | Mon | 0.9695 | 0.3345 | 0.1521 | 5.38% | −0.0997 | 0.3487 | 0.1646 | 5.46% | 0.0269 | 0.3064 | 0.1464 | 5.26% |
| | Tue | 0.7766 | 0.3518 | 0.6165 | 6.12% | 0.7794 | 0.3861 | 0.6251 | 6.24% | −0.0190 | 0.3051 | 0.6045 | 6.00% |
| | Wed | 0.8912 | 0.2614 | 0.1299 | 4.31% | 0.8900 | 0.2726 | 0.1354 | 4.52% | −0.0053 | 0.3059 | 0.1259 | 4.26% |
| | Thu | 0.8892 | 0.4786 | 0.4085 | 5.75% | 0.8892 | 0.4639 | 0.4054 | 5.71% | 0.0634 | 0.3074 | 0.4045 | 5.61% |
| | Fri | 0.7980 | 0.3227 | 0.2218 | 5.86% | −0.0205 | 0.3432 | 0.2342 | 5.97% | 0.0114 | 0.3111 | 0.1818 | 5.79% |
| | Sat | 0.9403 | 0.3747 | 0.2173 | 3.95% | 0.0646 | 0.2756 | 0.1158 | 3.87% | 0.0574 | 0.3112 | 0.1173 | 3.75% |
| | Sun | 0.9266 | 0.3388 | 0.2063 | 4.49% | −0.9567 | 0.3349 | 0.1936 | 4.45% | 0.0697 | 0.3187 | 0.1863 | 4.39% |

**Remark:** *To further demonstrate the comprehensive performance of the method at different sites, additional 10-min time series wind speed data were generated from four observation sites. Figure 5 and Table 4 show the forecasting results of Sites 1–4 in a week. But the differences are not great. In brief, it can be explained that SSA, which is able to denoise the time series wind speed data as a preprocessing method, and the APSOACO algorithm, which has better ability to detect optimizing weights, threshold, and bias value for BPNN, can improve forecasting accuracy. Furthermore, due to the vertical dataset selection method, the structure of the dataset is optimal, BPNN can reach the minimum error effectively.*

*4.4. Experiment III: The Performance of Linear Models and Nonlinear Models at Each Time Point*

In this experiment, four models including two nonlinear hybrid models (APSOACO-BPNN and DE-ELM) and two linear models (ARIMA and HW) are tested the performance of each time points among Monday Site 3.

(a)   The Monday every hours' results of Table 5 shows the following:

     (1)   SSA-APSOACO-BPNN obtained the lowest MAPE values of all single models at 1:00, 3:00, 10:00, 19:00 and 20:00 and the values are 3.57%, 3.05%, 3.84%, 3.38%, and 3.11%, respectively.

     (2)   SSA-DE-ELMNN achieved the lowest MAPE values of all single models at 2:00, 5:00, 6:00, 7:00, 9:00, 11:00, 12:00, 14:00, 15:00, 17:00, 2:00, and 21:00 to 24:00, and SSA-APSOACO-BPNN was also the model which obtained the best performance than others at whole hours.

     (3)   At 4:00, 8:00, 13:00, 16:00, and 18:00, SSA-HW reached the most accurate forecasting value.

     (4)   The single linear model: Although the single linear model ARIMA and the neuro-fuzzy networks ANFIS can achieve higher forecasting precision, their forecasting performance is worse than modified nonlinear at whole hour points.

     (5)   Compared with individual linear models, the modified nonlinear model performance better than single models and the MAPE value of the combined model has a significant improvement at whole hour points.

(b)   The whole Monday results of Figure 6 shows the following:

     (1)   Part A shows the MAPE values of five branch models, from the figure, ARIMA perform worst, the MAPE of other four models are approximately.

     (2)   From Figure 6B, the MSE and MAE of five models are not high, except 18:00 and 21:00 to 24:00. It indicates the single models perform about the same before 21:00.

     (3)   From Figure 6B, the forecasting results of ARIMA are closer than other models but not the whole hour points.

**Table 5.** The forecasting results of every hours of the four branch models in Site 3.

| Model | SSA-APSOACO-BPNN | | | SSA-DE-ELM | | | SSA-ARIMA | | | SSA-ANFIS | | | SSA-HW | | |
|---|---|---|---|---|---|---|---|---|---|---|---|---|---|---|---|
| Metric | MAE | MSE | MAPE | MAE | MSE | MAPE | MAE | MSE | MAPE | MAE | MSE | MAPE | MAE | MSE | MAPE |
| 1:00 | **0.1554** | **0.0419** | **3.57%** | 0.1787 | 0.0414 | 4.30% | 0.4303 | 0.2617 | 10.50% | 0.1680 | 0.0491 | 3.86% | 0.3250 | 0.1603 | 7.25% |
| 2:00 | 0.3047 | 0.1287 | 6.21% | 0.2553 | 0.1124 | **5.04%** | 0.3930 | 0.2446 | 7.77% | 0.3295 | 0.1505 | 6.72% | 0.4206 | 0.1851 | 8.92% |
| 3:00 | **0.1441** | **0.0518** | **3.05%** | 0.1699 | 0.0554 | 3.68% | 0.3270 | 0.1959 | 7.66% | 0.1559 | 0.0606 | 3.30% | 0.3106 | 0.1085 | 6.93% |
| 4:00 | 0.3856 | 0.2220 | 8.45% | 0.3279 | 0.1726 | 7.20% | 0.2943 | 0.1474 | 6.59% | 0.4170 | 0.2596 | 9.14% | **0.2403** | **0.0869** | **5.22%** |
| 5:00 | 0.2137 | 0.0681 | 3.98% | **0.1695** | **0.0465** | **3.02%** | 0.4883 | 0.3040 | 8.73% | 0.2311 | 0.0796 | 4.31% | 0.3041 | 0.1297 | 5.35% |
| 6:00 | 0.3141 | 0.1733 | 4.27% | **0.1495** | **0.0761** | **2.05%** | 0.4528 | 0.4138 | 6.29% | 0.3397 | 0.2027 | 4.62% | 0.3085 | 0.1105 | 4.68% |
| 7:00 | 0.3048 | 0.1259 | 4.09% | **0.2789** | **0.0906** | **3.81%** | 0.2842 | 0.1119 | 3.89% | 0.3296 | 0.1472 | 4.42% | 0.4090 | 0.2314 | 5.69% |
| 8:00 | 0.2738 | 0.0995 | 3.48% | 0.1935 | 0.0519 | 2.49% | 0.2234 | 0.0652 | 2.86% | 0.2961 | 0.1163 | 3.77% | **0.1056** | **0.0156** | **1.37%** |
| 9:00 | 0.2035 | 0.0566 | 3.44% | **0.1781** | **0.0640** | **3.11%** | 0.6844 | 0.5459 | 11.32% | 0.2200 | 0.0662 | 3.72% | 0.3208 | 0.1321 | 5.00% |
| 10:00 | **0.1829** | **0.0716** | **3.84%** | 0.2987 | 0.1057 | 6.28% | 0.4661 | 0.2433 | 9.80% | 0.1978 | 0.0838 | 4.15% | 0.3931 | 0.2007 | 7.95% |
| 11:00 | 0.3256 | 0.1719 | 6.21% | **0.3152** | **0.1547** | **6.01%** | 0.5971 | 0.5727 | 11.56% | 0.3522 | 0.2011 | 6.72% | 0.5763 | 0.3756 | 11.13% |
| 12:00 | 0.4616 | 0.4057 | 9.52% | **0.3711** | **0.3325** | **7.47%** | 0.5497 | 0.5517 | 13.13% | 0.4992 | 0.4745 | 10.30% | 0.3952 | 0.2066 | 9.08% |
| 13:00 | 0.2504 | 0.0881 | 5.16% | 0.3034 | 0.1114 | 6.40% | 0.4261 | 0.2483 | 8.75% | 0.2708 | 0.1030 | 5.58% | **0.2327** | **0.0789** | **4.89%** |
| 14:00 | 0.2360 | 0.1270 | 3.70% | **0.2217** | **0.1102** | **3.25%** | 0.5505 | 0.6382 | 8.54% | 0.2553 | 0.1486 | 4.00% | 0.4158 | 0.2431 | 6.62% |
| 15:00 | 0.3837 | 0.2613 | 6.39% | **0.2785** | **0.1224** | **4.80%** | 0.7279 | 0.7176 | 12.97% | 0.4149 | 0.3056 | 6.92% | 0.6729 | 0.5120 | 11.96% |
| 16:00 | 0.2643 | 0.0959 | 3.66% | 0.2562 | 0.0977 | 3.68% | 0.3107 | 0.2897 | 4.21% | 0.2858 | 0.1122 | 3.95% | **0.2623** | **0.0942** | **3.62%** |
| 17:00 | 0.3309 | 0.1321 | 4.37% | **0.2909** | **0.1327** | **3.90%** | 0.3205 | 0.1700 | 4.24% | 0.3578 | 0.1545 | 4.72% | 0.2667 | 0.1273 | 3.62% |
| 18:00 | 0.7540 | 0.6251 | 7.98% | 0.6813 | 0.5260 | 7.23% | 0.8699 | 0.8132 | 9.26% | 0.8154 | 0.7311 | 8.63% | **0.3306** | **0.1410** | **3.67%** |
| 19:00 | **0.3273** | **0.1796** | **3.38%** | 0.3465 | 0.1815 | 3.63% | 0.5828 | 0.4117 | 6.31% | 0.3540 | 0.2100 | 3.66% | 0.6051 | 0.4605 | 6.60% |
| 20:00 | **0.2372** | **0.1115** | **3.11%** | 0.2832 | 0.1519 | 3.70% | 0.4440 | 0.2433 | 5.63% | 0.2565 | 0.1304 | 3.36% | 0.3192 | 0.1341 | 4.00% |
| 21:00 | 0.5298 | 0.5733 | 6.31% | **0.5017** | **0.4639** | **6.00%** | 0.9227 | 1.5089 | 11.24% | 0.5729 | 0.6705 | 6.82% | 0.5247 | 0.6260 | 6.09% |
| 22:00 | 0.6814 | 0.5870 | 5.71% | **0.3377** | **0.1591** | **2.83%** | 1.1711 | 1.8294 | 9.91% | 0.7369 | 0.6865 | 6.18% | 0.8425 | 0.9869 | 7.24% |
| 23:00 | 0.5390 | 0.5112 | 4.25% | **0.5298** | **0.5283** | **4.17%** | 0.7180 | 1.0384 | 5.64% | 0.5829 | 0.5979 | 4.60% | 0.7315 | 0.6048 | 6.09% |
| 24:00 | 0.4266 | 0.2661 | 3.49% | **0.4149** | **0.2532** | **3.43%** | 0.5152 | 0.6012 | 4.33% | 0.4613 | 0.3112 | 3.78% | 0.4702 | 0.4084 | 4.01% |

Figure 6C also shows the 95% confidence intervals (CIs) obtained by five branch models, the figure indicates that both the upper and lower CI are close between three nonlinear models, but for linear models, they get larger CI which indicates the linear models reached worse results than nonlinear models.

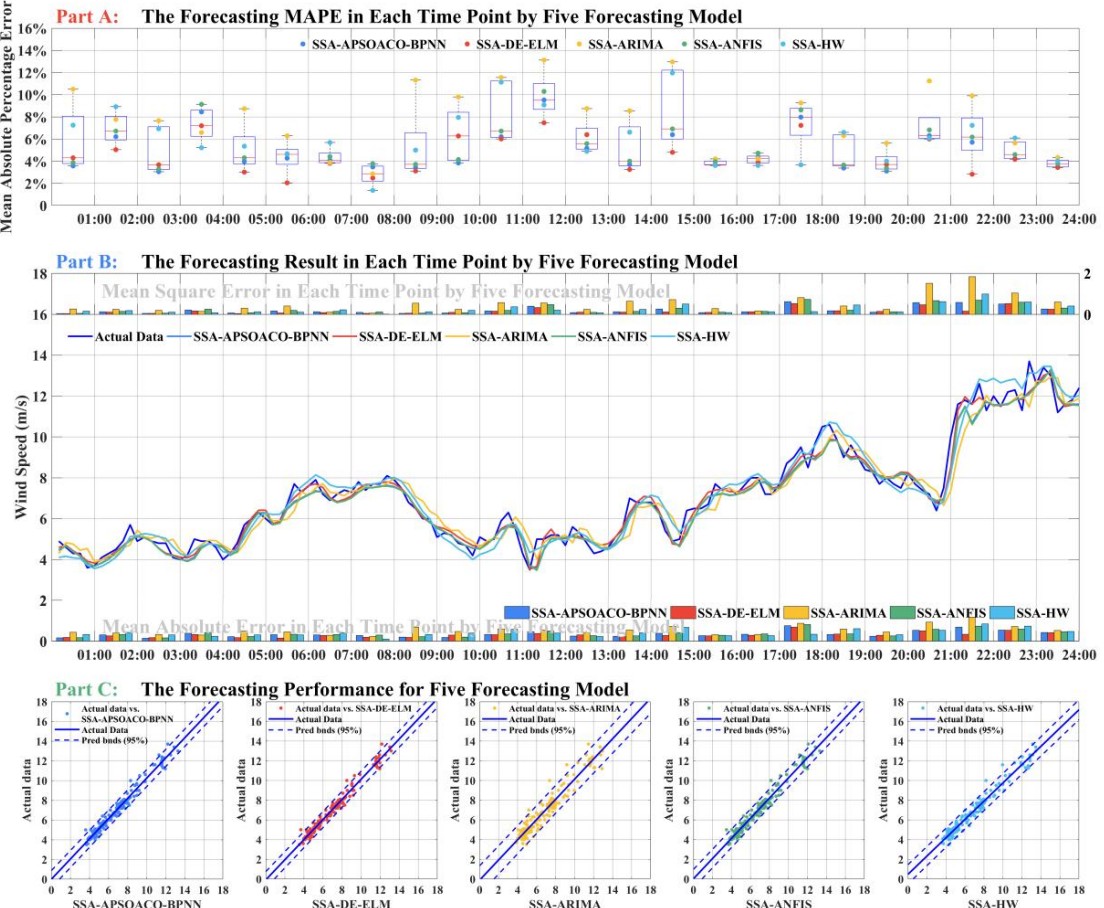

**Figure 6.** Typical forecasting performance of branch models of Monday in Site 3.

**Remark:** *The former experiment reveals there is no model that can reach best results at every time point; each model has advantages and disadvantages. The combined models can add up forecasting models to overcome these dilemmas. It has been regarded as an improvement on single models and an effective and simple way to improve forecasting stability. Understanding and predicting wind speed is very important for calculating wind farm and estimating the generation capacity and structural load of wind turbines [45]. By this study, it provides a scientific basis for wind turbine design, wind farm location and load reduction strategy.*

*4.5. Experiment IV: The Performance of Hybrid Models, Single Models and Combined Models at Four Sites*

As the results in Tables 6 and 7 shown, ANFIS is a special method in the development of neuro-fuzzy networks and shows good results in the modeling of nonlinear functions. ANFIS uses a hybrid algorithm of feed-forward network and least squares to adjust the premise parameters and conclusion parameters, and can automatically generate if–then rules that perform well on nonlinear work. In linear, nonstationarity, and in particular wind speed time series forecasting, ARIMA and HW are used in this experiment to judge whether these models fit the time series wind speed data or could forecast future points of the data.

Figure 7 shows the following:

(1) From Part A, our proposed combined model achieved the most accurate results compared with other models in these days because of the lower MAE values. It also shows that the forecasting of SSA-DE-ELMNN had better performance than other models, because the learning speed of ELMNN is extraordinarily fast. ELMNN had better generalization performance in feed-forward neural networks and tended to achieve solutions directly without these trivial problems [33].

(2) On these seven days, the results of SSA-ANFIS are nearly to SSA-DE-ELMNN which achieved the most accurate results compared with other models.

(3) The single linear model: Although ARIMA and HW can achieve higher prediction precision, prediction performance is worse than modified nonlinear and single nonlinear models.

(4) Compared with modified nonlinear models and individual nonlinear models, the MAPE value of the combined model had a significant improvement in forecasting accuracy.

(5) As Part C shows, the errors of combined model are very small, and our combined model also achieve a small CI.

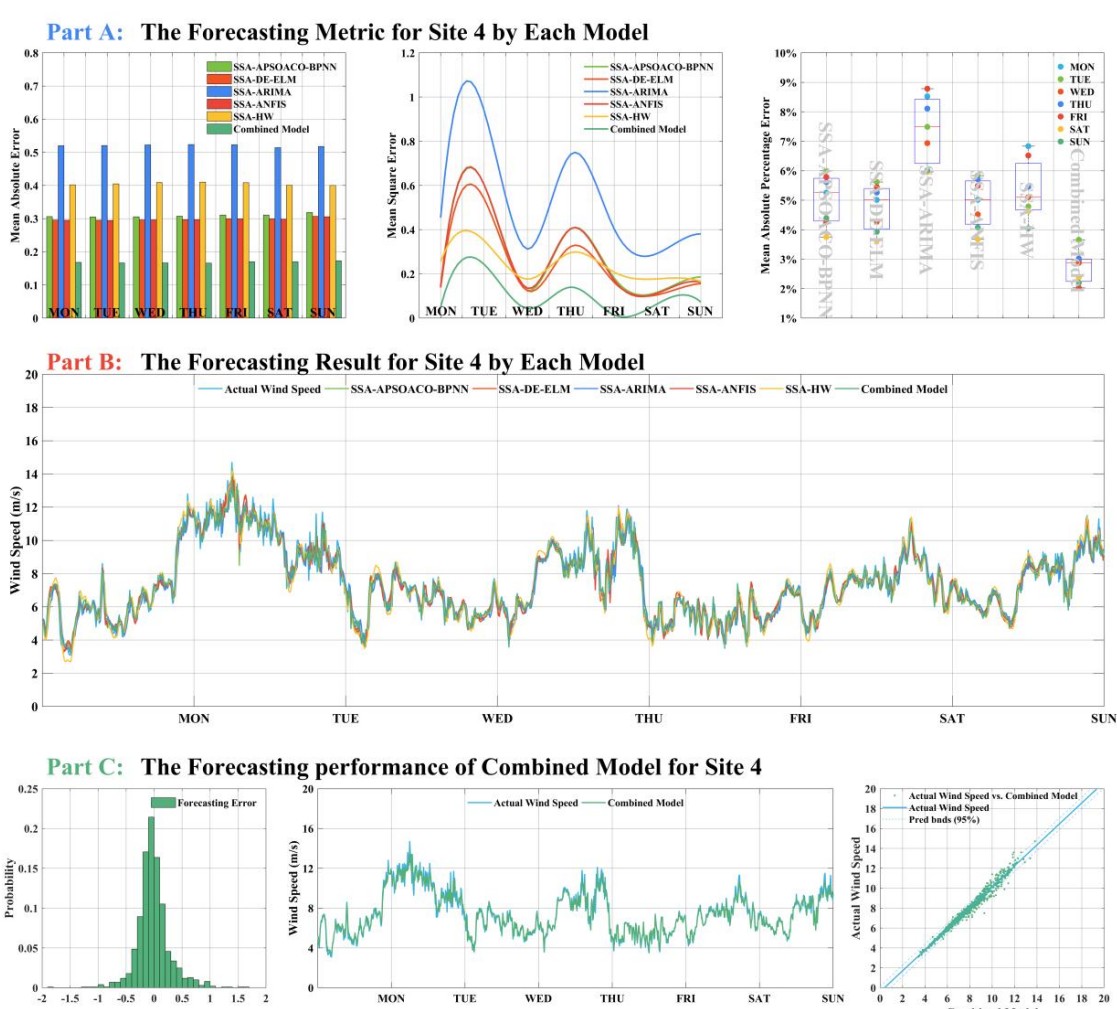

**Figure 7.** Typical forecasting performance of models in Site 4.

**Table 6.** Performance evaluation of different models for forecasts using the 10-min wind speed data in a week in Site 1 and 2.

| Time | Metric | Site 1 | | | | | | Site 2 | | | | | |
|---|---|---|---|---|---|---|---|---|---|---|---|---|---|
| | | SSA-APSOACO-BPNN | SSA-DE-ELMNN | SSA-ARIMA | SSA-ANFIS | SSA-HW | Combined Model | SSA-APSOACO-BPNN | SSA-DE-ELMNN | SSA-ARIMA | SSA-ANFIS | SSA-HW | Combined Model |
| Mon | AE | −0.0716 | −0.0617 | −0.0575 | 0.0461 | −0.0986 | −0.0617 | 0.9643 | 0.0977 | 0.0470 | −0.0580 | −0.0470 | 0.0508 |
| | MAE | 0.3612 | 0.3425 | 0.5564 | 0.3699 | 0.3425 | 0.1575 | 0.3434 | 0.3599 | 0.5493 | 0.3900 | 0.3434 | 0.1676 |
| | MSE | 0.2141 | 0.1935 | 0.5586 | 0.2215 | 0.1935 | 0.0418 | 0.1886 | 0.2042 | 0.5222 | 0.2764 | 0.1886 | 0.0573 |
| | MAPE | 6.08% | 5.85% | 9.37% | 6.20% | 5.85% | 2.66% | 5.00% | 5.16% | 7.81% | 5.42% | 5.00% | 2.26% |
| Tue | AE | 0.0118 | 0.0697 | 0.0332 | 0.0024 | 0.0255 | −0.0244 | 0.8253 | −0.0423 | −0.0259 | 0.0144 | −0.0063 | 0.0194 |
| | MAE | 0.3955 | 0.3832 | 0.5674 | 0.3993 | 0.3409 | 0.2322 | 0.4116 | 0.3816 | 0.5866 | 0.5538 | 0.3553 | 0.2555 |
| | MSE | 0.2763 | 0.2523 | 0.5096 | 0.2779 | 0.1844 | 0.0901 | 0.2663 | 0.2343 | 0.5453 | 0.4838 | 0.1874 | 0.1075 |
| | MAPE | 3.67% | 3.58% | 5.30% | 3.71% | 3.18% | 2.18% | 3.57% | 3.43% | 5.12% | 4.82% | 3.09% | 2.22% |
| Wed | AE | −0.0059 | −0.0005 | 0.0115 | −0.0039 | −0.0293 | 0.0567 | 0.8888 | 0.0574 | 0.0486 | 0.0155 | 0.0010 | 0.0996 |
| | MAE | 0.2421 | 0.4054 | 0.3717 | 0.2739 | 0.2450 | 0.1003 | 0.2570 | 0.3449 | 0.3782 | 0.3696 | 0.3637 | 0.2301 |
| | MSE | 0.0938 | 0.2868 | 0.2343 | 0.1150 | 0.0953 | 0.0148 | 0.1023 | 0.1964 | 0.2465 | 0.2298 | 0.1974 | 0.0972 |
| | MAPE | 4.62% | 3.76% | 6.70% | 4.92% | 4.43% | 1.94% | 3.99% | 5.17% | 5.76% | 5.56% | 5.50% | 3.36% |
| Thu | AE | 0.0050 | 0.0526 | −0.0328 | 0.0245 | 0.0259 | 0.0364 | −0.0502 | −0.0784 | −0.0656 | −0.0601 | −0.0058 | −0.1167 |
| | MAE | 0.3613 | 0.2637 | 0.5329 | 0.3782 | 0.3592 | 0.1673 | 0.3737 | 0.4945 | 0.5427 | 0.5603 | 0.4647 | 0.3539 |
| | MSE | 0.2186 | 0.1065 | 0.5144 | 0.2346 | 0.2263 | 0.0546 | 0.2171 | 0.4236 | 0.5635 | 0.5413 | 0.3811 | 0.2033 |
| | MAPE | 4.90% | 4.87% | 7.06% | 5.14% | 4.82% | 2.12% | 3.98% | 5.42% | 5.91% | 6.06% | 4.97% | 3.87% |
| Fri | AE | −0.0151 | −0.0189 | 0.0046 | 0.0612 | −0.0343 | −0.0093 | 0.8571 | 0.0521 | −0.0497 | −0.0661 | −0.0730 | 0.0444 |
| | MAE | 0.2914 | 0.3020 | 0.4534 | 0.3341 | 0.2995 | 0.1310 | 0.2721 | 0.3021 | 0.4195 | 0.4151 | 0.3498 | 0.0819 |
| | MSE | 0.1356 | 0.1442 | 0.3333 | 0.1835 | 0.1440 | 0.0341 | 0.1192 | 0.1592 | 0.3058 | 0.2908 | 0.2024 | 0.0095 |
| | MAPE | 5.55% | 5.77% | 8.67% | 6.51% | 5.73% | 2.58% | 4.79% | 5.20% | 7.30% | 7.40% | 6.12% | 1.50% |
| Sat | AE | 0.0164 | 0.0383 | −0.0391 | −0.0423 | −0.0424 | 0.0383 | 0.0125 | −0.0037 | 0.0391 | 0.0423 | 0.0024 | −0.0383 |
| | MAE | 0.2565 | 0.2472 | 0.3821 | 0.3237 | 0.2472 | 0.0874 | 0.2774 | 0.2344 | 0.4176 | 0.3093 | 0.3174 | 0.1109 |
| | MSE | 0.1079 | 0.1007 | 0.2484 | 0.1602 | 0.1007 | 0.0134 | 0.1212 | 0.1012 | 0.2992 | 0.1544 | 0.1712 | 0.0257 |
| | MAPE | 4.21% | 4.05% | 6.11% | 5.04% | 4.05% | 1.42% | 3.65% | 3.35% | 5.52% | 4.01% | 4.25% | 1.39% |
| Sun | AE | −0.0564 | −0.0269 | −0.0495 | −0.0651 | −0.0596 | 0.0366 | −0.0032 | −0.0265 | 0.0577 | 0.0636 | 0.0147 | −0.0149 |
| | MAE | 0.2432 | 0.2339 | 0.3307 | 0.2433 | 0.2256 | 0.1415 | 0.2357 | 0.2567 | 0.4812 | 0.4582 | 0.4741 | 0.1727 |
| | MSE | 0.0939 | 0.0897 | 0.1808 | 0.0938 | 0.0899 | 0.0353 | 0.0952 | 0.1252 | 0.3603 | 0.3244 | 0.3422 | 0.0737 |
| | MAPE | 3.21% | 3.12% | 4.39% | 3.21% | 3.00% | 1.76% | 2.75% | 2.98% | 5.65% | 5.36% | 5.55% | 2.23% |

**Table 7.** Performance evaluation of different models for forecasts using the 10-min wind speed data in a week in Site 3 and 4.

| Time | Metric | Site 3 | | | | | | Site 4 | | | | | |
|---|---|---|---|---|---|---|---|---|---|---|---|---|---|
| | | SSA-APSOACO-BPNN | SSA-DE-ELMNN | SSA-ARIMA | SSA-ANFIS | SSA-HW | Combined Model | SSA-APSOACO-BPNN | SSA-DE-ELMNN | SSA-ARIMA | SSA-ANFIS | SSA-HW | Combined Model |
| Mon | AE | −0.1005 | 0.0650 | 0.0616 | −0.0251 | −0.0386 | 0.5469 | 0.0269 | 0.0117 | 0.1047 | −0.0562 | −0.0086 | −0.0259 |
| | MAE | 0.3367 | 0.3055 | 0.5313 | 0.4076 | 0.3709 | 0.1945 | 0.3064 | 0.2958 | 0.5200 | 0.2951 | 0.4019 | 0.1680 |
| | MSE | 0.2094 | 0.1726 | 0.5070 | 0.2650 | 0.2522 | 0.0751 | 0.1464 | 0.1384 | 0.4545 | 0.1426 | 0.2558 | 0.0502 |
| | MAPE | 4.84% | 4.48% | 7.96% | 6.12% | 5.30% | 2.88% | 5.26% | 5.01% | 8.53% | 5.02% | 6.84% | 2.97% |
| Tue | AE | 0.7546 | 0.0222 | −0.0302 | −0.0177 | 0.2880 | 0.0168 | −0.0190 | −0.0424 | 0.0403 | 0.0773 | −0.0571 | 0.0089 |
| | MAE | 0.5235 | 0.5092 | 0.6674 | 1.4148 | 0.3842 | 0.3270 | 0.3051 | 0.2949 | 0.5205 | 0.2945 | 0.4048 | 0.1667 |
| | MSE | 0.4094 | 0.3872 | 0.6632 | 3.1693 | 0.2141 | 0.1600 | 0.6045 | 0.5411 | 0.9400 | 0.6050 | 0.3511 | 0.2437 |
| | MAPE | 4.52% | 4.41% | 5.85% | 11.94% | 3.35% | 2.86% | 6.00% | 5.62% | 7.49% | 5.84% | 4.79% | 3.66% |
| Wed | AE | 0.0014 | −0.0144 | −0.0189 | −0.0280 | 0.3539 | 0.0136 | −0.0053 | 0.0128 | −0.0263 | −0.0993 | −0.0829 | 0.0242 |
| | MAE | 0.2539 | 0.2513 | 0.3721 | 0.6860 | 0.2683 | 0.0609 | 0.3059 | 0.2963 | 0.5226 | 0.2964 | 0.4092 | 0.1666 |
| | MSE | 0.0964 | 0.0943 | 0.2283 | 0.7185 | 0.1122 | 0.0060 | 0.1259 | 0.1237 | 0.3122 | 0.1354 | 0.1767 | 0.0433 |
| | MAPE | 3.96% | 3.92% | 5.76% | 10.56% | 4.10% | 0.95% | 4.26% | 4.27% | 6.94% | 4.52% | 5.11% | 2.88% |
| Thu | AE | 0.9218 | 0.0028 | 0.0199 | 0.0357 | −0.0058 | −0.0230 | 0.0634 | 0.0222 | 0.0191 | 0.0330 | 0.0472 | −0.0173 |
| | MAE | 0.3609 | 0.3870 | 0.5024 | 0.3839 | 0.3292 | 0.2533 | 0.3074 | 0.2971 | 0.5235 | 0.2970 | 0.4103 | 0.1657 |
| | MSE | 0.2434 | 0.2507 | 0.4999 | 0.2633 | 0.2015 | 0.1439 | 0.4045 | 0.3243 | 0.7413 | 0.4054 | 0.2968 | 0.1402 |
| | MAPE | 3.87% | 4.15% | 5.36% | 4.07% | 3.49% | 2.49% | 5.61% | 5.26% | 8.11% | 5.71% | 5.46% | 3.02% |
| Fri | AE | 0.8143 | 0.0185 | −0.0169 | −0.0693 | −0.0029 | 0.0427 | 0.0114 | −0.0150 | −0.0127 | −0.0768 | 0.0209 | 0.0103 |
| | MAE | 0.3007 | 0.2989 | 0.4575 | 0.3007 | 0.3516 | 0.1186 | 0.3111 | 0.2997 | 0.5232 | 0.2993 | 0.4086 | 0.1702 |
| | MSE | 0.1629 | 0.1552 | 0.3920 | 0.1629 | 0.2140 | 0.0242 | 0.1818 | 0.1623 | 0.4040 | 0.1715 | 0.2080 | 0.0142 |
| | MAPE | 5.14% | 5.12% | 7.85% | 5.14% | 6.08% | 2.16% | 5.79% | 5.45% | 8.78% | 5.49% | 6.52% | 2.03% |
| Sat | AE | 0.8858 | −0.0768 | 0.0266 | 0.0065 | −0.0046 | −0.0022 | 0.0574 | 0.0148 | 0.0457 | 0.0489 | −0.0224 | 0.0113 |
| | MAE | 0.3091 | 0.3016 | 0.4275 | 0.3091 | 0.3436 | 0.1319 | 0.3112 | 0.2996 | 0.5142 | 0.2978 | 0.4007 | 0.1702 |
| | MSE | 0.1807 | 0.1606 | 0.3478 | 0.1807 | 0.2017 | 0.0355 | 0.1173 | 0.1056 | 0.2933 | 0.1129 | 0.1776 | 0.0652 |
| | MAPE | 4.44% | 4.26% | 6.01% | 4.35% | 4.86% | 1.84% | 3.75% | 3.62% | 5.96% | 3.68% | 4.63% | 2.38% |
| Sun | AE | 0.9516 | 0.0165 | 0.0528 | 0.0649 | 0.0147 | −0.0240 | 0.0697 | 0.0414 | 0.0285 | 0.0244 | 0.0131 | −0.0219 |
| | MAE | 0.2787 | 0.2683 | 0.4113 | 0.2727 | 0.2729 | 0.1833 | 0.3187 | 0.3071 | 0.5177 | 0.3060 | 0.4001 | 0.1732 |
| | MSE | 0.1237 | 0.1204 | 0.2719 | 0.1237 | 0.1312 | 0.0769 | 0.1863 | 0.1565 | 0.3811 | 0.1627 | 0.1656 | 0.0729 |
| | MAPE | 3.33% | 3.16% | 4.79% | 3.23% | 3.28% | 2.05% | 4.39% | 3.93% | 6.02% | 4.08% | 4.05% | 2.20% |

It is further shown in Tables 6 and 7 that the performance of wind speed forecasting improved by our proposed combined model with SDCS. The wind speed forecasting performance in a week at Sites 1–4 was applied as experimental data. It is obvious that the forecasting performance of our proposed combined model is more accurate than the other single hybrid model for each site. This result responds to the reliability and stability of our proposed combined model taking into account the random nature of the wind and its spatiotemporal variation. For Sites 1–4, the SSA-DE-ELMNN is always superior to any other single hybrid forecasting model. For example, the SSA-DE-ELMNN approach overmatched the other single hybrid models in a week forecasting with a lower MAPE (4.48%) than SSA-APSOACO-BPNN, SSA-ARIMA, SSA-ANFIS, and SSA-HW models with value 4.84%, 7.96%, 6.12%, and 5.30% in Monday from site 3. In addition, our proposed combined model improves the MAPE with the value 0.11% at least (Thursday in site 2) and it also has higher accuracy and stability in wind speed forecasting.

**Remark:** *Our proposed combined model is more effective than the hybrid linear or nonlinear single model in forecasting wind speed at both site. The minimum performance metrics were obtained by the combined models.*

### 4.6. Experiment V: Test of SDCS

The ability of CS and SDCS to solve the test functions and global optimal solution of multiple test functions is evaluated in this section. The test functions and parameters [46] of the algorithm are listed in Tables 8 and 9, respectively.

**Table 8.** Test functions.

| Function Name | Test Function | Variable Domain | Global Optimum |
| :---: | :---: | :---: | :---: |
| Sphere | $f(x) = x^2 + y^2$ | $[-5.12, 5.12]$ | $f_{min}(0, 0, \ldots, 0) = 0$ |
| Rosenbrock | $f(x) = 100 \times (y - x^2)^2 + (1 - x)^2$ | $[-2.084, 2.084]$ | $f_{min}(1, 1, \ldots, 1) = 0$ |
| Rastrigin | $f(x) = \sum_{i=1}^{d} (x_i^2 - 10(2\pi x_i) + 10)$ | $[-5.12, 5.12]$ | $f_{min}(0, 0, \ldots, 0) = 0$ |
| Schaffer | $f(x) = \frac{sin^2 \sqrt{\sum_{i=1}^{d} x_i^2} - 0.5}{1 + 0.001 \left( \sum_{i=1}^{d} x_i^2 \right)}$ | $[-5.12, 5.12]$ | $f_{min}(0, 0, \ldots, 0) = 0$ |

**Table 9.** Experimental parameters of modified model SDCS.

| Experimental Parameters | CS | SDCS |
| :---: | :---: | :---: |
| Num Cuckoos | 50 | 50 |
| Min Number Of Eggs | 20 | 20 |
| Max Number Of Eggs | 40 | 40 |
| Max Iter | 20,000 | 20000 |
| Knn Cluster Num | 1 | 1 |
| Motion Coeff | 20 | 20 |
| Accuracy | $1.00 \times 10^{-10}$ | $1.00 \times 10^{-10}$ |
| Max Num Of Cuckoos | 20 | 20 |
| Radius Coeff | 0.05 | 0.05 |
| Cuckoo Pop Variance | $1.00 \times 10^{-10}$ | $1.00 \times 10^{-10}$ |

The results of this experiment are as follows:

For the sphere function, both CS and SDCS successfully obtain convergence.

(1)　The max, min, and average values of the iterations are 298, 144, and 12 for CS, but for SDCS, max, min, and average values are only 12, 8, and 7.8 when the dimension is 10.

(2)　When the dimension is 20, the max, min, and average values of the iterations of CS are 389, 314, and 347. However, for SDCS, max, min, and average values are only 29, 18, and 22.8.

(3) When the dimension is 50, the max, min, and average values of the iterations of CS are 597, 558, and 576.4. However, for SDCS, max, min, and average values are only 183, 147, and 166, respectively.

For the Rosenbrock function, CS cannot obtain the convergence when the dimension of variables is 2, whereas the max, min, and average values of the iterations are 185, 99, and 173 for SDCS.

For the Rastrigin function, both algorithms can successfully obtain convergence except CS when the dimension is 50. The performance of SDCS is better than CS. Both algorithms can successfully obtain convergence when the dimensions are 10 and 20; when the dimension is 50, SDCS can achieve optimized results, but CS cannot.

For the Rosenbrock function, both algorithms can successfully achieve optimized results, and the performance of SDCS is better than CS. See Table 10.

**Table 10.** The experimental parameters of SDCS.

| Test Function | Dimension | Algorithm | Max Value of Iteration | Min Value of Iteration | Average Value of Iteration | Convergence Rate |
|---|---|---|---|---|---|---|
| Sphere | 10 | CS | 298 | 144 | 201 | 1 |
| | | SDCS | 12 | 8 | 10.8 | 1 |
| | 20 | CS | 389 | 314 | 347 | 1 |
| | | SDCS | 29 | 18 | 22.8 | 1 |
| | 50 | CS | 597 | 558 | 576.4 | 1 |
| | | SDCS | 183 | 147 | 166 | 1 |
| Rosenbrock | 2 | CS | - | - | - | - |
| | | SDCS | 185 | 99 | 173 | 1 |
| Rastrigin | 10 | CS | 461 | 375 | 389 | 1 |
| | | SDCS | 197 | 164 | 178 | 1 |
| | 20 | CS | 968 | 781 | 815 | 0.89 |
| | | SDCS | 628 | 478 | 569 | 1 |
| | 50 | CS | - | - | - | - |
| | | SDCS | 1542 | 973 | 1328 | 0.93 |
| Schaffer | 2 | CS | 1326 | 1071 | 1315 | 0.83 |
| | | SDCS | 76 | 64 | 69 | 1 |

**Remark:** *Above all, the iteration of SDCS is less than CS and when CS can not achieve the SDCS also obtained the goal. From this experiment, the performance of SDCS is better than CS.*

*4.7. Summary*

From former four experiments, we found the following facts:

(1) Experiment I and Table 5 indicate that wind speed data is both linear and nonlinear by hypothesis test and the wind speed data cannot be considered as linear or nonlinear. So the linear models and nonlinear models considered in our proposed forecasting model is correct and necessary.

(2) The results from Experiment II, Figure 4 and Table 3 reveal that SSA-APSOACO-BPNN performed better than SSA-APSO-BPNN and SSA-ACO-BPNN in accuracy.

(3) Because the DE algorithm has better capacity to search optimizing weights and threshold value for ELMNN, SSA-DE-ELMNN is more stable and accurate than the single model SSA-ELMNN as the Experiment II, Figure 5 and Table 4 show.

(4) As shown in Figure 6 and Table 5 from Experiment III, different branch models can obtain the best results at different time points. This character caters to the feature of combined theory, which can avoid the loss of information in fitting a single model, reduce the randomness, and improve forecasting precision. Our proposed combined model can provide more accurate results than the single models. From the results, the minimum performance metrics were obtained by the combined models.

(5) Experiment IV, our proposed combined model provided more accurate results than the single models. From the results shown in Figure 7 and Tables 6 and 7, the minimum performance metrics were obtained by the combined models.

(6) From the last experiment V, it shows that the optimizing performance of SDCS is better than that of CS, which means SDCS can optimize the weight coefficients of the combined model more effective.

In addition, the single models also improved in this paper, and these modified single models provided more accurate results than the single models as shown in Experiment II. As shown in the above experiments, the proposed combined model, which achieved higher accuracy, possesses a more powerful forecasting ability than the benchmark models. Improved wind speed forecasting would be extremely significant to the energy grid and wind farms. With the integration of large-scale wind power into the power grid, the safety and stability of the grid would face severe challenges. Accurate forecasting of wind power generation (wind speed) is an effective way to enhance the security and controllability of the power grid and to realize reasonable dispatch of the power grid. Wind speed forecasting and wind power generation forecasting have received attention around the world.

## 5. Conclusions

Wind speed forecasting is of great significance to the operation of wind farms in terms of economy and safety. Accurate and reliable forecasting results have a significant impact on wind farms, which in turn have an influence on the economy. In this study, the preprocessing technique reduces the uncertainty and irregularity of wind speed data, and effectively improves the performance of wind speed forecasting. Furthermore, to improve the capacity of our proposed combined forecasting model, we integrated the improved cuckoo search algorithm and developed a new algorithm named SDCS. Our proposed combined model, which includes SSA-APSOACO-BPNN, SSA-DE-ELMNN, SSA-ARIMA, SSA-ANFIS, and SSA-HW, is more effective than the hybrid nonlinear or linear single model in forecasting wind speed based on the above reasons, it improves the MAPE with the value range from 0.11% (Site 2, compared with ASS-APSOACO-BPNN) to 9.61% (Site 3, compared with SSA-ANFIS) and it also has higher accuracy and stability in wind speed forecasting. As the results show, our proposed model is more accurate than compared models. So, according to our research, the combined model can be used in wind farms to save operating costs and wind power. By improving forecasting accuracy and stability, the combined model also can be used to predict wind speed and power dispatch, resulting in various benefits such as avoiding grid collapse and saving economic dispatch.

**Author Contributions:** Y.L. carried on programming and writing of the whole manuscript; S.Z. carried on the validation and visualization of experiment results; X.C. and J.W. provided the overall guide of conceptualization and methodology.

**Funding:** The research was funded by Gansu science and technology program "Study on the forecasting methods of very short-term wind speeds" (Grant number: 1506RJZA187).

**Conflicts of Interest:** The authors declare no conflict of interest.

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
