# Peer review of "Artificial Combined Model Based on Hybrid Nonlinear Neural Network Models and Statistics Linear Models—Research and Application for Wind Speed Forecasting"

_sustainability, doi:10.3390/su10124601_

Round 1

Reviewer 1 Report

1.     The authors have made a good attempt to write a paper addressing a study on artificial combined model based on hybrid nonlinear neural network models and statistics linear models - research and application for wind speed forecasting.

2.     This paper is very well crafted and is a great improvement from the first version.

3.     Even though the conclusion is quite extensive, the conclusion should go deeper but concise and the important numerical results of the work should be given. In the conclusion section, the author should be objective and specific, presenting the main results from the study. What are the most sounding quantifiable results of the study?

4.     The authors should avoid citations in the conclusion.

Author Response

1. The authors have made a good attempt to write a paper addressing a study on artificial combined model based on hybrid nonlinear neural network models and statistics linear models - research and application for wind speed forecasting.

Respond: Thanks for your suggestions.

2. This paper is very well crafted and is a great improvement from the first version.

Respond: Thanks for your suggestions.

3. Even though the conclusion is quite extensive, the conclusion should go deeper but concise and the important numerical results of the work should be given. In the conclusion section, the author should be objective and specific, presenting the main results from the study. What are the most sounding quantifiable results of the study?

Respond: Thanks for your suggestions. Conclusion was truncated for the sake brevity, deeper and more focused on our work. The most sounding quantifiable results was shown in the conclusion (line 662-677 and we marked them in yellow).

4. The authors should avoid citations in the conclusion.

Respond: Thanks for your suggestions. We have deleted the citation.

Reviewer 2 Report

The article's quality has been increased in the revised version and the authors  have tried to address my concerns, therefore I feel the paper is suitable for publication.

Author Response

The article's quality has been increased in the revised version and the authors  have tried to address my concerns, therefore I feel the paper is suitable for publication.

Respons: Thanks for your comments and suggestions.
